# VECO: VEctor COnformity Based OOD Detection in Text and Multimodal Models

**Mouïn Ben Ammar**[◇,†,*] **, Arturo Mendoza**[†]**, Antoine Manzanera**[◇]**, Gianni Franchi**[◇]
**U2IS Lab ENSTA Paris**[◇]**, Palaiseau, FRANCE**
**Safran Tech**[†]**, Chateaufort 78117, FRANCE**
`{first.last}@ensta.fr`[◇]`, safrangroup.com`[†]

**Reviewed on OpenReview:** `https://openreview.net/forum?id=sMbGqh7Zvt&noteId=TnplKIBVNh`

## Abstract

Out-of-distribution (OOD) detection is critical for the reliable deployment of natural language processing and multimodal document understanding systems, where domain and semantic shifts are unavoidable. While many post-hoc OOD detection methods were developed for vision models, their direct transfer to textual and multimodal Transformer architectures remains poorly understood. We show that, unlike in vision benchmarks, feature-space provides the dominant OOD signal for text and document models, consistently outperforming logit-based and hybrid detectors. Building on this observation, we introduce **VECO** (*VEctor COnformity*), a geometry-aware, purely feature-based OOD scoring framework that implements a stable soft contrast between in-distribution conformity and residual-space deviation. We instantiate VECO using principal-subspace conformity for multimodal document models and Mahalanobis distance conformity for text classifiers, reflecting modality-aligned representation structure. VECO achieves state-of-the-art and consistent performance improvements on multimodal document and text classification benchmarks. These results highlight the modality-dependent nature of OOD detection and the importance of adapting score design to representation cues.

## 1 Introduction

Out-of-distribution (OOD) detection is a central component of reliable machine learning systems, enabling models to recognize inputs that deviate from their training distribution and to abstain or trigger fallback mechanisms accordingly. While significant progress has been made in vision-based OOD detection, similar reliability concerns arise in *natural language processing (NLP)* and *multimodal document understanding* systems deployed in real-world, high-stakes settings. In these domains, distribution shift is inevitable, arising from changes in topic, writing style, document layout, or visual structure. Despite their practical importance, OOD detection for textual and multimodal Transformer models remains comparatively under-explored (Hendrycks et al., 2020; Lang et al., 2023).

Recent studies suggest that classical confidence- and feature-based OOD detectors developed for computer vision can transfer reasonably well to NLP and multimodal tasks (Baran et al., 2023; Constantinou et al., 2024), especially when applied to modern pretrained architectures (Hendrycks et al., 2020). Many of these approaches aim to *improve the separability of in-distribution (ID) and OOD representations* prior to scoring (Zhou et al., 2021; Constantinou et al., 2024; Darrin et al., 2024), for instance via contrastive learning, attention masking, or multi-layer representation aggregation.

Despite these efforts, most existing approaches focus on reshaping representations to increase separability, while comparatively less attention has been paid to whether *post-hoc OOD scoring functions* are well adapted to the characteristics of textual and multimodal representations. Linguistic and multimodal representations possess structural properties that might differ from those learned in vision tasks. Transformers such as

BERT exhibit strong anisotropy (Ethayarajh, 2019) and highly correlated subspaces (Gao et al., 2021), while multimodal models encode text, layout, and visual cues in intertwined ways. These properties can strongly influence post-hoc OOD scores; consequently, directly transferring OOD detectors designed for vision models may be suboptimal.

In this work, we investigate how *feature-space* characteristic and geometry influence OOD detection in textual and multimodal Transformer models, and how this geometry can be exploited to design more stable post-hoc scoring functions. We evaluate two representative settings: (i) multimodal document classification using LayoutLMv3 (Huang et al., 2022) fine-tuned on the FinanceDocs (Constantinou et al., 2024) and Tobacco3482 (Harley et al., 2015) datasets, and (ii) text classification using BERT (Devlin et al., 2019) fine-tuned on SST-2 (Socher et al., 2013), evaluated under both near- and far-OOD shifts.

Our empirical analysis reveals that *feature-based* scores consistently outperform logit-based and hybrid methods (combining both logit and feature signals) on these benchmarks. Notably, hybrid scores such as ViM (Wang et al., 2022) and NECO (Ammar et al., 2024) underperform relative to their own feature-only components, in contrast to their typical behavior on vision benchmarks (Liang et al., 2023). This suggests that the textual and multimodal feature-space already provides strong OOD separability cues, and that vision-inspired hybridization schemes can degrade performance rather than provide complementary signal in these settings.

Motivated by these observations, we propose lightweight, geometry-aware adaptations of feature-based OOD scoring. Our main contributions are:

- We conduct a systematic evaluation of post-hoc OOD detection on textual and multimodal Transformer models, showing that feature-space provides the strongest and most stable OOD signal across both modalities.

- We introduce **VECO** (*VEctor COnformity*), a geometry-aware, purely feature-based OOD scoring framework that implements a stable soft contrast between ID conformity and residual-space deviation. We further provide a likelihood-inspired motivation under a simple block-structured Gaussian model, offering intuition for why such contrasts are effective.

- We instantiate VECO in two principled variants—**VECO-P**, based on principal-subspace conformity for multimodal document models, and **VECO-M**, based on Mahalanobis conformity for text classifiers, showing that modality-aligned conformity signals yield state-of-the-art performance in these settings.

Our analysis focuses on frozen, pretrained models fine-tuned for classification and on sentence- and document-level tasks. Within this scope, the results highlight that *OOD detection is inherently modality dependent*, and that score designs accounting for the structure of the underlying representation space yield consistent improvements.

## 2 Background

While OOD detection has been extensively studied in vision, recent studies have explored adapting post-hoc OOD detection techniques from vision to textual and multimodal Transformer models. A central difficulty in this transfer is that Transformer representations exhibit geometric structure that may differs substantially from visual features: language models such as BERT produce highly anisotropic embeddings with a small number of dominant directions (Ethayarajh, 2019; Mu et al., 2017), and multimodal document models encode textual, spatial, and visual information in tightly coupled subspaces (Huang et al., 2022). These properties can strongly affect the behavior of classical post-hoc OOD scores, making direct transfer from vision settings potentially suboptimal. Baran et al. (2023) systematically evaluated classical post-hoc OOD detectors—such as maximum softmax probability (MSP) (Hendrycks & Gimpel, 2017), energy-based scores (Liu et al., 2020b), and Mahalanobis distance (Lee et al., 2018), and showed that many degrade substantially under realistic linguistic shifts.

Most prior works in NLP and document-level OOD detection addresses this challenge by *reshaping representations* to improve ID/OOD separability. Contrastive objectives such as SimCSE (Gao et al., 2021) encourage more isotropic sentence embeddings, and Zhou et al. (2021) showed that such training improves robustness to unseen domains. Other approaches aggregate representations across layers to capture complementary syntactic and semantic cues, either by constructing holistic embeddings (Chen et al., 2022) or by multi-layer embeddings aggregation without retraining (Xu et al., 2021).

A second line of work operates at the level of *scoring functions or architectural adaptations.* Darrin et al. (2024) aggregated OOD scores across Transformer layers to improve stability. In multimodal document understanding, additional complexity arises from the interaction of text, layout, and visual structure. Constantinou et al. (2024) proposed attention-head masking for LayoutLMv3 (Huang et al., 2022) to reshape multimodal representations and improve ID/OOD separation. Related work in visual question answering explored training auxiliary modules for selective prediction abstention under uncertainty (Whitehead et al., 2022).

Overall, existing work in NLP and document-level OOD detection has focused primarily on manipulating representations through contrastive learning, masking, or aggregation. Comparatively little attention paid to the design of *post-hoc scoring functions* that explicitly account for the geometric structure of textual and multimodal embeddings. Our work addresses this gap by studying geometry-driven OOD scores tailored for these settings.

## 3 Methodology and Proposed Scoring Functions

### 3.1 Why modality-aware feature scores?

Hybrid OOD detectors that mix feature- and logit-derived signals (e.g., ViM (Wang et al., 2022) and NECO (Ammar et al., 2024)) are often more effective than single-source OOD scores in vision benchmarks. However, in our textual and multimodal Transformer settings, we consistently observe a different trend: feature-based scores (e.g., Mahalanobis distance, residual norms, and the NECO feature component) outperform both logit-based and hybrid scores, with the latter often underperforming their own feature-only components. This aligns with previously observed results (Baran et al., 2023; Chen et al., 2022; Constantinou et al., 2024), and suggests the features provide the dominant separability cue in these settings, while standard logit-feature scores coupling can empirically degrade performance rather than provide complementary information.

Motivated by this, we introduce **VECO** (*VEctor COnformity*), a feature-based OOD score that implements a smooth contrast between two complementary geometric cues: *ID conformity* and *residual-space deviation*, which OOD samples tend to activate more strongly. VECO avoids dependence on logits, which proved unstable when combined with features in our textual and multimodal settings. We further provide theoretical insight via a simple block-structured Gaussian model in a PCA basis, under which the optimal log-likelihood ratio depends on the same principal–residual contrast. VECO can thus be viewed as a smooth surrogate that captures the same principal–residual contrast highlighted by equation 5. We instantiate this principle in two variants: **VECO-P**, based on principal-subspace conformity, and **VECO-M**, based on Mahalanobis conformity.

### 3.2 VECO: a soft-contrast score in feature space

Our empirical analysis across textual and multimodal benchmarks shows that feature-space geometry alone provides a stronger and more stable OOD signal than logit-based or hybrid combinations in these settings. This observation motivates a feature-based score, more adapted to multimodal document and text classification settings. We formalize **VECO** as a purely feature-based score that measures how strongly a representation conforms to the dominant in-distribution signal geometry.

Concretely, let $\mathbf{f} \in \mathbb{R}^d$ denote the feature vector of a test sample. Let $\mathbf{U}_k \in \mathbb{R}^{d \times k}$ denote the top-$k$ principal components computed by PCA on ID training features, and let $\mathbf{U}_r \in \mathbb{R}^{d \times m}$ denote a set of $m$ low-variance PCA directions (a *residual subspace*), with $k + m \leq d$. [1]

$$s_p = \|\mathbf{U}_k^\top \mathbf{f}\|_2^2, \qquad s_r = \|\mathbf{U}_r^\top \mathbf{f}\|_2^2. \tag{1}$$

Empirically, ID representations tend to concentrate along the principal subspace (large $s_p$) (Ammar et al., 2024) while remaining small in the residual directions (small $s_r$) (Wang et al., 2022), whereas OOD samples typically exhibit the reverse trend. VECO combines these two geometric signals into a single smooth contrasting score that increases with principal-space conformity and decreases with residual-space deviation.

**Score definition.** We define VECO as a soft contrast between the principal- and residual- subspace magnitudes:

$$S_{\text{VECO}}(\mathbf{f}) = s_p - \log\big(e^{s_p} + e^{\alpha s_r}\big), \tag{2}$$

where $\alpha$ is a scale calibration factor estimated from the ID training set:

$$\alpha = \frac{\sum_{i=1}^n \|U_k^\top \mathbf{f}_i^{\text{train}}\|_2^2}{\sum_{i=1}^n \|U_r^\top \mathbf{f}_i^{\text{train}}\|_2^2}, \tag{3}$$

Here, $\{\mathbf{f}_i^{\text{train}}\}_{i=1}^n$ denote the in-distribution training features. The subspaces $U_k$ and $U_r$ are computed once from the training set and remain fixed at test time. Thus, $\alpha$ serves as a calibration constant that normalizes the relative magnitudes of principal and residual energies, and is determined using an empirical average over the training data and requires no additional tuning. This normalization ensures that both conformity and deviation signals to contribute on comparable scales to the soft contrast.

Empirically, $\mathbf{U}_r$ is fixed using the lowest $m = 512$ eigenvectors of the training covariance matrix across experiment. The value of $k$ (dimension of the principal subspace) is selected per benchmark, as it depends on the learned ID structure.

This form can be read as a two-way soft competition between an ID-conformity term ($s_p$) and a residual-deviation term ($\alpha s_r$), yielding higher scores for ID samples. It acts as a smooth surrogate of the contrast suggested by the likelihood-ratio structure discussed in Appendix B. Broadly, VECO acts as a feature-space conformity measure that implicitly integrates both NECO-like (principal structure) (Ammar et al., 2024) and ViM-like (residual deviation) (Wang et al., 2022) cues without relying on classifier outputs.

Algorithm 1 summarizes the computations.

### 3.3 A probabilistic motivation for VECO

We provide a simple generative view that motivates VECO as a smooth surrogate to a log-likelihood ratio based on the contrast between principal-space conformity and residual-space deviation. The following analysis is intended as a conceptual motivation rather than a generative model of the feature distribution, and serves to highlight the underlying representation's geometric structure.

To provide theoretical intuition for VECO, we introduce a deliberately simplified model of ID and OOD features expressed in the PCA basis of the ID training data means, isolating the dominant geometric contrast observed empirically. This abstraction yields a closed-form log–likelihood ratio. Its dependence on the principal and residual magnitudes ($s_p, s_r$) highlights the same structural contrast promoted by VECO.

**Idealized generative model.** We decompose the feature vector as

$$\mathbf{z}_p = \mathbf{U}_k^\top \mathbf{f}, \qquad \mathbf{z}_r = \mathbf{U}_r^\top \mathbf{f}, \tag{4}$$

---

[1]In practice, $m$ is chosen as a fixed fraction of the lowest-variance components rather than the full orthogonal complement.

where $\mathbf{U}_k$ and $\mathbf{U}_r$ are defined in Section 3.2, and their corresponding magnitudes $s_p$ and $s_r$ are specified in equation 1.

A class-marginalized Gaussian model consistent with the geometric structure of ID features (in a PCA-aligned basis) assumes

$$\text{ID: } \mathbf{z}_p \sim \mathcal{N}(\mathbf{0},\, \sigma_p^2 \mathbf{I}_k + \Sigma_{\text{disc}}), \qquad \mathbf{z}_r \sim \mathcal{N}(\mathbf{0},\, \sigma_r^2 \mathbf{I}_m),$$

$$\text{OOD: } \mathbf{z}_p \sim \mathcal{N}(\mathbf{0},\, \tilde{\sigma}_p^2 \mathbf{I}_k), \qquad \mathbf{z}_r \sim \mathcal{N}(\mathbf{0},\, \tilde{\sigma}_r^2 \mathbf{I}_m),$$

2

Here $\Sigma_{\text{disc}}$ captures class-dependent variance in the principal subspace in a basis that we assume aligns with the (class-mean) discriminative subspace, while within-class isotropic variance remains small. We adopt an idealized assumption that the class-discriminative covariance $\Sigma_{\text{disc}}$ is diagonal in this basis. In particular, letting $C$ denote the number of in-distribution classes, we define

$$\Sigma_{\text{disc}} = \text{diag}(\lambda_1, \ldots, \lambda_C, 0, \ldots, 0) \in \mathbb{R}^{k \times k}, \quad \lambda_i \geq 0,$$

so that only the first $C$ principal directions exhibit additional class-discriminative variance, while the remaining $k - C$ directions contain no such structure. This formulation reflects a class-conditional variability confined to a low-dimensional subset of principal directions, as can be seen in Figure 1. This abstraction is consistent with Neural Collapse-like geometry observations in (Ammar et al., 2024; Papyan et al., 2020). To corroborate the geometric assumptions underlying our model, we visualize ID class structure and OOD dispersion in both high-variance and low-variance PCA directions (Figure 2). In the principal subspace (PC1–PC2), ID features cluster tightly around their class means, while OOD features show no alignment with these discriminative directions. In contrast, projections onto high-index components (e.g., PC100–PC101) show that ID features remain tightly concentrated near the origin, whereas OOD features display substantially larger variance. Together, these observations empirically support the block-structured idealization and motivate contrasting principal conformity with residual deviation.

**Log-likelihood ratio (structural form).** Under this Gaussian model, the class-marginal log–likelihood ratio $\log \frac{p(\mathbf{f}|\text{ID})}{p(\mathbf{f}|\text{OOD})}$ can be written as

$$\Lambda(f) \;\propto\; \frac{1}{2}\,\bar{w}\, s_p \;-\; \frac{1}{2}\,\gamma_r\, s_r, \qquad \bar{w}, \gamma_r > 0. \tag{5}$$

so the induced ranking is a monotone function of the pair $(s_p, s_r)$. This structure motivates OOD scores that reward a principal-space conformity signal ($s_p$ large) and penalize residual deviation ($s_r$ large), as implemented by the contrast of these two terms in VECO.

A detailed derivation is provided in Appendix B.

**Remark 1.** *While $\Lambda(\mathbf{f})$ is expressed in terms of the full feature $\mathbf{f}$, the simplified model depends only on the projected coordinates $(\mathbf{z}_p, \mathbf{z}_r)$. Intermediate directions orthogonal to both subspaces are not modeled explicitly and are treated as approximately shared noise that does not affect the induced ordering. Focusing on $(\mathbf{z}_p, \mathbf{z}_r)$ thus isolates the subspaces that appear most informative for distinguishing ID from OOD, consistent with our goal of deriving a feature–based surrogate rather than a full generative model in the feature space.*

**Remark 2.** *In the Gaussian analysis, the coefficients $w_i$ arise from differences in inverse variances (precisions) along the principal directions of the in-distribution covariance. Under the assumed Gaussian model, these coefficients take the form $w_i = \frac{1}{\tilde{\sigma}_p^2} - \frac{1}{\sigma_p^2 + \lambda_i}$, We approximate that the leading eigenvalues satisfy $\lambda_i \approx \lambda$ for $i \in [1, C]$, allowing the coefficients $w_i$ to be approximated by a single scalar $\bar{w}$. The full derivation and discussion of this approximation are provided in Appendix B. Similarly, the residual coefficient is given by $\gamma_r = \frac{1}{\sigma_r^2} - \frac{1}{\tilde{\sigma}_r^2}$.*

---

[2]We denote the dimensionality of the residual subspace by $m$ to avoid notational confusion with the residual variance $\sigma_r^2$; $m$ corresponds to the number of retained low-variance PCA directions.

*These coefficients determine the relative contribution of principal and residual energies in the idealized likelihood ratio. VECO does not attempt to approximate these coefficients directly, but instead preserves the ordering induced by the contrast between principal conformity and residual deviation.*

**VECO as a smooth contrast.** VECO does not attempt to approximate the likelihood itself, but instead implements a smooth contrast that depends on the same two geometric quantities $(s_p, s_r)$. The VECO score is strictly increasing in $s_p$ and strictly decreasing in $s_r$, the same monotonic directions highlighted by the structural form of the likelihood ratio $\Lambda(\mathbf{f})$.

VECO is defined in equation 2 as a smooth contrast of $(s_p, s_r)$ via a two-way softmax, while calibrating their relative scales, which is equivalent to the negative log of a two-way softmax over $\{s_p, \alpha s_r\}$:

$$S_{\text{VECO}}(\mathbf{f}) = -\log\big(1 + e^{\alpha s_r - s_p}\big). \tag{6}$$

with $\alpha$ defined in equation 3, see Appendix B.3 for proof. This surrogate increases when principal-space conformity increases and residual deviation decreases. Figure 9 shows the distribution of $s_p$, $s_r$ and VECO, illustrating how ID samples concentrate at high $s_p$ and low $s_r$, OOD samples exhibiting substantially larger residual magnitudes, and VECO contrasting these two informative geometric signals as a smooth reshaping function.

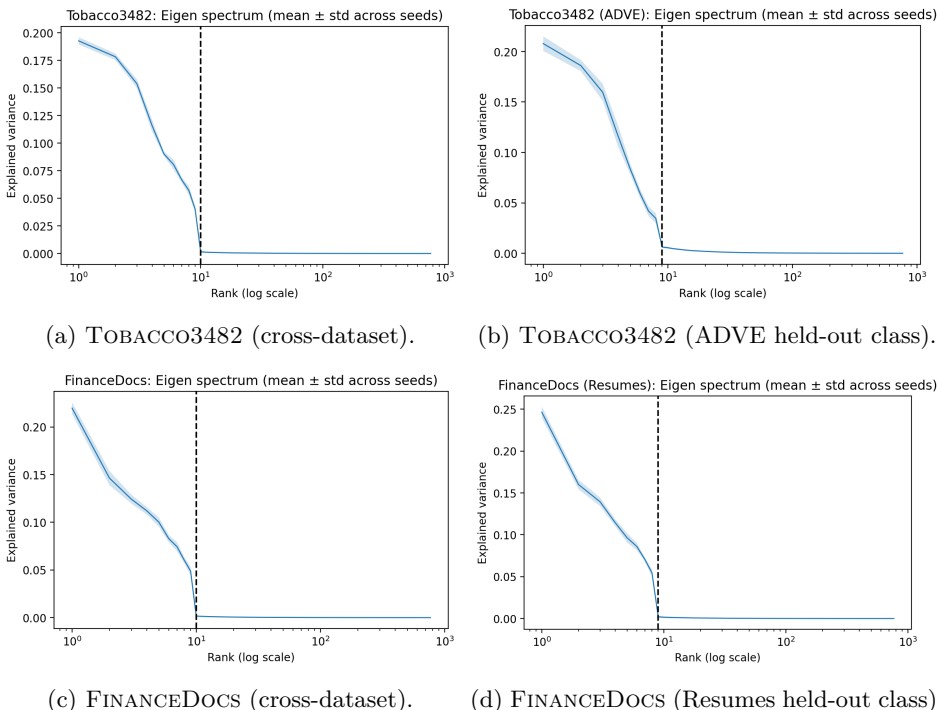

(a) TOBACCO3482 (cross-dataset). (b) TOBACCO3482 (ADVE held-out class).

(c) FINANCEDOCS (cross-dataset). (d) FINANCEDOCS (Resumes held-out class).

Figure 1: Eigenvalue spectra of LayoutLMv3 penultimate-layer features across all multimodal in-distribution settings. The vertical dashed line marks the number of ID classes.

### 3.4 Toy verification of the likelihood contrast

To further corroborate the geometric validity of the VECO soft-contrast, we construct a controlled toy setting in which the idealized block-Gaussian assumptions from Section 3.3 hold by design. In particular, we generate synthetic ID and OOD features following our idealized setting.

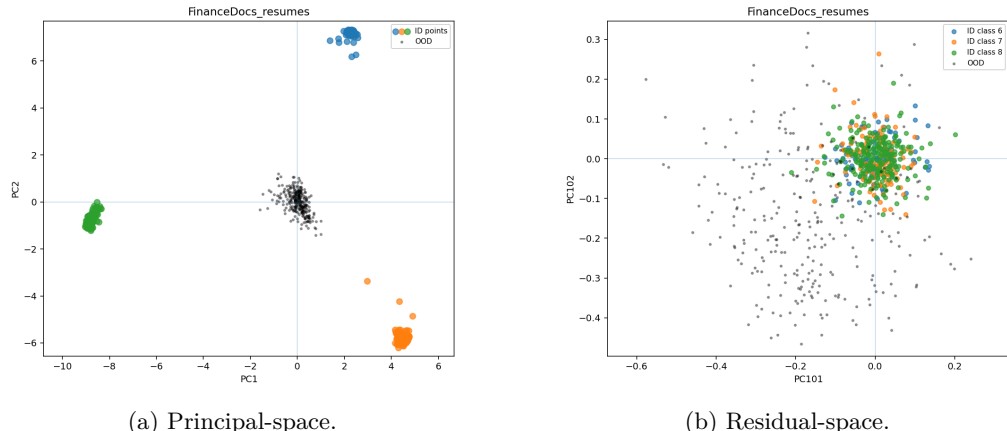

(a) Principal-space.                                                   (b) Residual-space.

Figure 2: PCA projections of penultimate-layer features for the FinanceDocs (Resumes held-out class) case. (*Left*) Projection onto the top principal components (PC1–PC2). (*Right*) Projection onto low-variance components (PC100–PC101). For clarity, we visualize three random ID classes (in colors), while OOD samples are shown in gray.

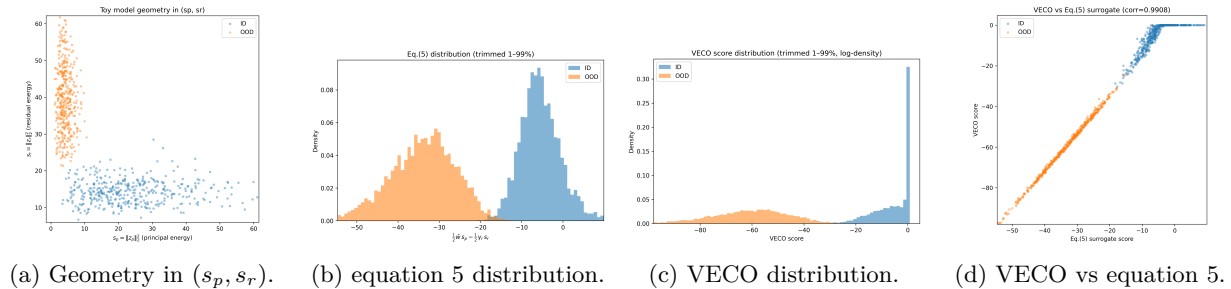

(a) Geometry in $(s_p, s_r)$.   (b) equation 5 distribution.   (c) VECO distribution.   (d) VECO vs equation 5.

Figure 3: **Toy verification of the likelihood contrast behind VECO.** (a) ID and OOD separate primarily in residual energy: OOD samples exhibit larger $s_r = \|z_r\|_2^2$, while ID exhibits comparatively small $s_r$ and higher principal energy $s_p = \|z_p\|_2^2$ due to the discriminative variance injected into the principal block.

We compare equation 5 ID/OOD separation to the VECO score. Figure 3 shows that both scores separate ID and OOD in the same direction and exhibit a strong monotonic relationship over the generated samples, supporting VECO as a smooth surrogate of the principal–residual likelihood contrast.

## 3.5 Instantiations: VECO-P and VECO-M

VECO follows a soft-contrast template that combines a chosen *ID-conformity* signal with a residual-deviation term:

$$\text{(ID conformity)} \; - \; \log\bigl(\exp(\text{ID conformity}) + \exp(\text{residual deviation})\bigr), \tag{7}$$

The two signals are derived from orthogonal subspaces and can be used independently to identify OOD effectively, suggesting that a proper combination is likely to increase OOD separability. We consider two representative instantiations of this principle, differing only in the choice of the ID-conformity term.

**VECO-P (principal-space conformity).** For multimodal document models, we use $s_p(\mathbf{f}) = \|\boldsymbol{U}_k^\top \mathbf{f}\|_2^2$ as in equation 2, yielding the VECO-P score.

**VECO-M (Mahalanobis-distance conformity).** For text encoders, we opt for a Mahalanobis conformity signal. Let $(\mu, \Sigma)$ be the empirical mean and covariance of ID training features. Define

$$s_{\mathrm{Mah}}(\mathbf{f}) = -(\mathbf{f} - \mu)^{\top} \Sigma^{-1} (\mathbf{f} - \mu), \tag{8}$$

and replace $s_p(\mathbf{f})$ by $s_{\mathrm{Mah}}(\mathbf{f})$ in equation 2:

$$S_{\mathrm{VECO\text{-}M}}(\mathbf{f}) = s_{\mathrm{Mah}}(\mathbf{f}) - \log\Big(e^{s_{\mathrm{Mah}}(\mathbf{f})} + e^{\alpha\, s_r(\mathbf{f})}\Big). \tag{9}$$

This preserves the same soft-contrast structure while using covariance-aware conformity, which can be advantageous for binary text classification benchmarks.

We report both variants across modalities; empirically, VECO-P is most effective on our document classification benchmarks, while VECO-M is most effective for far-OOD detection on BERT features.

## 4 Experimental Setting

### 4.1 OOD detection task

In all experiments, out-of-distribution detection is formulated as a binary discrimination problem between in-distribution (ID) and OOD samples using post-hoc scores computed on frozen model representations. Given a test input, an OOD scoring function assigns a scalar confidence value, with higher scores indicating greater conformity to the ID distribution. Detection performance is evaluated by thresholding these scores, without modifying model parameters or retraining.

To study post-hoc OOD detection across different modalities, we consider two representative settings: (i) text classification with pretrained contextual language models, and (ii) multimodal document classification combining textual, visual, and spatial cues. This section describes the benchmarks, training procedures, and evaluation protocols used in our experiments.

### 4.2 Benchmarks

**Near vs. Far OOD.** Following common practice in NLP, we distinguish between *near-OOD* inputs, which exhibit stylistic or background shift within related semantic domain, and *far-OOD* inputs, where both the task and underlying semantics differ substantially from ID data. This distinction reflects realistic deployment scenarios in which models encounter both mild and severe forms of distribution shift (Arora et al., 2021).

**Text classification.** For text classification, we fine-tune **BERT-base (uncased)** (Devlin et al., 2019) on the **SST-2** sentiment classification task (Socher et al., 2013). We evaluate two types of distributional shift following Arora et al. (2021):

- **Near-OOD:** IMDB (Maas et al., 2011) and YELP POLARITY (Zhang et al., 2015), which share similar vocabulary with SST-2 but differ in document lengths, writing styles, and source domains.
- **Far-OOD:** AG-NEWS (Zhang et al., 2015) (topic classification), RTE (Giampiccolo et al., 2007) (textual entailment), and TREC-QC (Voorhees & Tice, 2000) (question classification), introducing both semantic and task-level shift.

This setup is consistent with prior NLP OOD detection studies (Baran et al., 2023; Chen et al., 2022). We refer to the penultimate-layer `[CLS]` embeddings as features.

**Multimodal document classification.** For multimodal document classification, we follow the protocol of Constantinou et al. (2024) and evaluate on the FINANCEDOCS and TOBACCO3482 datasets (Harley et al., 2015). Both datasets consist of scanned business or administrative documents with OCR text, spatial layout information, and visual features.

We evaluate two complementary OOD settings:

(i) **Held-out-class (intra-dataset) OOD**, where one document class is excluded from training and treated as OOD (e.g., *Advertisement* for Tobacco3482 and *Resumes* for FinanceDocs), probing semantic shift within a fixed document domain.

(ii) **Cross-dataset OOD**, where models trained on one dataset are evaluated on the other, testing semantic shift involving different layouts, topics, and visual structure.

For clarity and consistency with the results tables, we use the following naming conventions:

- **FinanceDocs**: ID = all FinanceDocs classes; OOD = Tobacco3482.

- **FinanceDocs_Resumes**: ID = FinanceDocs excluding *Resumes*; OOD = held-out *Resumes*.

- **Tobacco3482**: ID = all Tobacco3482 classes; OOD = FinanceDocs.

- **Tobacco3482_ADVE**: ID = Tobacco3482 excluding *Advertisement*; OOD = held-out *Advertisement*.

### 4.3 Models and Evaluation Protocol

In all experiments, models are fine-tuned for classification and then frozen; all OOD scores are computed post-hoc on fixed representations.

**BERT-base.** We fine-tune `bert-base-uncased` using cross-entropy loss and AdamW with standard hyperparameters for three epochs. The checkpoint with the best validation accuracy is retained, achieving approximately 93% validation accuracy, consistent with standard fine-tuning performance.

**LayoutLMv3.** For multimodal document classification, we use LayoutLMv3 (Huang et al., 2022), a Transformer-based architecture that jointly encodes textual, visual, and spatial layout information. The model is fine-tuned separately on each dataset: 15 epochs for FINANCEDOCS (achieving $\sim 99\%$ ID accuracy) and 50 epochs for TOBACCO3482 (achieving $\sim 90\%$). In all experiments, we use the checkpoint with the best validation accuracy.

**Baselines Methods.** We compare VECO against a broad set of standard post-hoc OOD detectors. *Logit-based* baselines include MSP (Hendrycks & Gimpel, 2017), Energy (Liu et al., 2020b), GradNorm (Huang et al., 2021), KL-Matching and MaxLogit (Hendrycks et al., 2022). *Feature-based* methods include Mahalanobis (Lee et al., 2018), residual (Wang et al., 2022), NaN (Park et al., 2023), and the feature-norm component of NECO (Ammar et al., 2024), referred to as NECO-Feature. *Hybrid* approaches that combine logits and features include ViM (Wang et al., 2022), ASH variants (Djurisic et al., 2022), Energy+ReAct (Sun et al., 2021), and NECO (Ammar et al., 2024). All methods are evaluated in a strictly post-hoc setting without additional training, adaptations, or access to OOD data.

**Evaluation metrics.** We evaluate OOD detection performance using the *area under the ROC curve* (AUROC) and the *false positive rate at 95% true positive rate* (FPR95). AUROCmeasures overall separability between in-distribution (ID) and OOD samples, while FPR95captures performance in the low–false-negative regime that is most relevant for deployment. Both metrics are reported as percentages, and results are reported as mean $\pm$ standard deviation over ten random seeds to account for fine-tuning variability.

## 5 Results

We evaluate the proposed VECO scoring framework and its two instantiations, VECO-P and VECO-M, on multimodal LayoutLMv3 document benchmarks and the BERT-based SST-2 text benchmark, using the experimental setups in Section 4. All methods are evaluated post-hoc on frozen, fine-tuned encoders, and we analyze each modality separately for clarity. We further evaluate the sensitivity of VECO to its hyperparameters. Results in Appendix C show that performance is stable across a broad range of values for $k$ and $m$, while the calibration factor $\alpha$ improves stability across datasets.

## 5.1 Multimodal Document Classification (LayoutLMv3)

Tables 1 and 2 report OOD detection performance on Tobacco3482 and FinanceDocs datasets.

Table 1: **OOD detection performance on the Tobacco3482 benchmark.** Each method is evaluated in two settings: (i) *cross-dataset OOD*, and (ii) *held-out-class OOD* (Tobacco3482_ADVE), AUROCand FPR95are reported (mean ± std over 10 seeds). VECO-P achieves the strongest performance in both settings.

| Method | Tobacco3482 | | Tobacco3482 (ADVE) | | Average | |
|---|---|---|---|---|---|---|
| | AUROC | FPR95 | AUROC | FPR95 | AUROC | FPR95 |
| MSP | 75.27 ± 3.88 | 79.26 ± 3.11 | 86.89 ± 3.49 | 58.74 ± 13.36 | 81.08 ± 3.69 | 69.00 ± 8.24 |
| MaxLogit | 75.10 ± 6.37 | 76.19 ± 5.59 | 87.91 ± 3.80 | 52.39 ± 12.36 | 81.51 ± 5.14 | 64.29 ± 8.97 |
| KL-Matching | 77.15 ± 2.16 | 80.63 ± 3.22 | 86.60 ± 2.97 | 57.48 ± 10.83 | 81.88 ± 2.57 | 69.05 ± 7.02 |
| GradNorm | 75.54 ± 6.67 | 73.16 ± 9.44 | 90.23 ± 3.55 | 41.00 ± 14.78 | 82.89 ± 5.11 | 57.08 ± 12.10 |
| Energy | 75.61 ± 7.25 | 73.33 ± 7.78 | 88.92 ± 4.41 | 45.52 ± 13.32 | 82.26 ± 5.81 | 59.43 ± 10.55 |
| Energy+React | 74.71 ± 7.54 | 75.08 ± 8.05 | 87.04 ± 6.29 | 48.13 ± 14.76 | 80.87 ± 6.93 | 61.60 ± 11.40 |
| ASH-B | 58.02 ± 17.83 | 86.70 ± 10.67 | 56.94 ± 23.71 | 82.43 ± 17.62 | 57.48 ± 20.77 | 84.57 ± 14.14 |
| ASH-P | 75.61 ± 7.25 | 73.33 ± 7.78 | 88.92 ± 4.41 | 45.52 ± 13.32 | 82.26 ± 5.83 | 59.43 ± 10.55 |
| ASH-S | 75.22 ± 6.50 | 75.24 ± 6.32 | 88.39 ± 3.78 | 49.04 ± 11.91 | 81.81 ± 5.14 | 62.14 ± 9.11 |
| Mahalanobis | 82.34 ± 2.54 | 73.64 ± 7.02 | 94.91 ± 0.76 | 24.87 ± 5.93 | 88.63 ± 1.65 | 49.26 ± 6.48 |
| NECO-Feature | 81.61 ± 2.51 | 74.71 ± 5.14 | 93.39 ± 0.87 | 32.52 ± 7.04 | 87.50 ± 1.69 | 53.62 ± 6.09 |
| NECO | 76.25 ± 5.66 | 75.65 ± 4.91 | 88.52 ± 3.50 | 50.35 ± 11.71 | 82.39 ± 4.58 | 63.00 ± 8.31 |
| NaN | 78.08 ± 5.09 | 70.72 ± 12.08 | 92.65 ± 1.52 | 32.74 ± 7.42 | 85.37 ± 10.31 | 51.73 ± 26.86 |
| Residual | 82.28 ± 2.50 | 74.70 ± 5.98 | 94.64 ± 0.83 | 25.22 ± 5.98 | 88.46 ± 1.67 | 49.96 ± 5.98 |
| ViM | 82.19 ± 2.47 | 75.13 ± 5.40 | 94.70 ± 0.76 | 24.70 ± 5.71 | 88.44 ± 1.62 | 49.91 ± 5.56 |
| VECO-M (ours) | 82.24 ± 2.51 | 74.21 ± 5.70 | 94.88 ± 0.77 | 25.35 ± 5.86 | 88.56 ± 8.93 | 49.78 ± 5.55 |
| **VECO-P (ours)** | **84.51 ± 2.75** | **70.02 ± 7.46** | **95.79 ± 0.65** | **17.78 ± 2.94** | **90.15 ± 1.70** | **43.90 ± 5.20** |

Table 2: **OOD detection performance on the FinanceDocs benchmark.** We evaluate two settings: (i) *cross-dataset OOD*, and (ii) *held-out-class OOD* (FinanceDocs_Resumes), AUROCand FPR95are reported (mean ± std over 10 seeds). VECO-P matches or exceeds all baselines, achieving the lowest average FPR95.

| Method | FinanceDocs | | FinanceDocs (Resumes) | | Average | |
|---|---|---|---|---|---|---|
| | AUROC | FPR95 | AUROC | FPR95 | AUROC | FPR95 |
| MSP | 95.12 ± 1.38 | 17.73 ± 4.00 | 93.99 ± 6.18 | 19.78 ± 13.35 | 94.56 ± 3.78 | 18.75 ± 8.68 |
| MaxLogit | 94.79 ± 1.30 | 20.05 ± 5.70 | 92.52 ± 8.39 | 23.34 ± 19.44 | 93.65 ± 4.85 | 21.70 ± 12.57 |
| KL-Matching | 96.88 ± 0.63 | 14.47 ± 2.71 | 97.96 ± 1.03 | 9.41 ± 8.49 | 97.42 ± 0.76 | 11.94 ± 5.60 |
| GradNorm | 94.60 ± 1.67 | 20.97 ± 8.74 | 92.82 ± 8.58 | 23.04 ± 17.65 | 93.71 ± 5.13 | 22.00 ± 13.20 |
| Energy | 94.65 ± 1.45 | 21.47 ± 7.99 | 92.13 ± 9.03 | 24.97 ± 21.95 | 93.39 ± 5.24 | 23.22 ± 14.97 |
| Energy+React | 94.49 ± 1.65 | 21.17 ± 6.95 | 92.15 ± 8.46 | 25.90 ± 21.58 | 93.32 ± 5.06 | 23.53 ± 14.27 |
| ASH-B | 65.85 ± 25.39 | 73.78 ± 23.59 | 74.86 ± 20.96 | 65.93 ± 28.15 | 70.36 ± 23.18 | 69.85 ± 25.87 |
| ASH-P | 94.65 ± 1.45 | 21.47 ± 7.99 | 92.13 ± 9.03 | 24.97 ± 21.95 | 93.39 ± 5.24 | 23.22 ± 14.97 |
| ASH-S | 94.81 ± 1.31 | 20.11 ± 5.84 | 92.54 ± 8.39 | 23.31 ± 19.48 | 93.68 ± 4.85 | 21.71 ± 12.66 |
| Mahalanobis | 98.49 ± 0.40 | 6.48 ± 1.87 | 99.16 ± 0.39 | 0.66 ± 1.11 | 98.83 ± 0.48 | 3.57 ± 1.49 |
| NECO-Feature | 98.43 ± 0.41 | 7.71 ± 2.71 | 99.05 ± 0.48 | 2.01 ± 2.57 | 98.74 ± 0.44 | 4.86 ± 2.64 |
| NECO | 96.06 ± 1.10 | 15.42 ± 4.61 | 95.07 ± 5.00 | 15.87 ± 12.56 | 95.56 ± 3.05 | 15.65 ± 8.59 |
| NaN | 93.26 ± 2.21 | 31.78 ± 10.22 | 93.62 ± 5.93 | 25.44 ± 19.84 | 93.44 ± 0.25 | 28.61 ± 4.49 |
| Residual | **98.67 ± 0.37** | 6.26 ± 1.88 | 99.27 ± 0.38 | 0.81 ± 1.33 | 98.97 ± 0.42 | 3.54 ± 1.61 |
| ViM | 98.66 ± 0.36 | 6.29 ± 1.84 | 99.25 ± 0.39 | 0.89 ± 1.66 | 98.96 ± 0.42 | 3.59 ± 1.75 |
| VECO-M (ours) | 98.65 ± 0.37 | 6.09 ± 1.91 | 99.28 ± 0.39 | 0.62 ± 1.00 | 98.97 ± 0.45 | 3.35 ± 3.87 |
| **VECO-P (ours)** | 98.63 ± 0.37 | **5.70 ± 1.77** | **99.31 ± 0.37** | **0.39 ± 0.68** | **98.97 ± 0.37** | **3.04 ± 1.23** |

Table 3: **Near-OOD detection on SST-2.** ID = SST-2; OOD = IMDB and Yelp Reviews. AUROCand FPR95are reported (mean ± std over 10 seeds).

| Method | Near OOD | | | | | |
| | IMDB | | Yelp reviews | | Average | |
| | AUROC | FPR95 | AUROC | FPR95 | AUROC | FPR95 |
| --- | --- | --- | --- | --- | --- | --- |
| **MSP** | 63.39 ± 1.23 | 89.51 ± 1.25 | 62.30 ± 1.99 | 90.97 ± 1.29 | 62.84±0.77 | 90.24 ± 1.03 |
| **MaxLogit** | 63.03 ± 2.38 | 88.65 ± 1.13 | 62.46 ± 3.05 | 89.84 ± 1.60 | 62.75±0.40 | 89.24 ± 0.84 |
| **KL-Matching** | 49.28 ± 1.76 | 89.27 ± 1.37 | 45.74 ± 2.01 | 90.79 ± 1.39 | 47.51±2.50 | 90.03 ± 1.07 |
| **GradNorm** | 48.70 ± 5.97 | 94.16 ± 1.80 | 48.31 ± 10.69 | 93.98 ± 2.39 | 48.51±0.27 | 94.07 ± 0.13 |
| **Energy** | 63.06 ± 2.45 | 88.30 ± 2.36 | 62.53 ± 3.10 | 89.27 ± 2.40 | 62.79±0.38 | 88.78 ± 0.69 |
| **Energy+React** | 52.61 ± 11.33 | 93.00 ± 7.45 | 50.35 ± 11.20 | 95.12 ± 6.56 | 51.48±1.59 | 94.06 ± 1.50 |
| **ASH-B** | 50.59 ± 7.16 | 97.69 ± 1.46 | 49.89 ± 6.59 | 96.27 ± 3.57 | 50.24±0.49 | 96.98 ± 1.00 |
| **ASH-P** | 51.06 ± 11.15 | 93.95 ± 6.93 | 49.08 ± 12.12 | 95.90 ± 5.13 | 50.07±1.40 | 94.92 ± 1.38 |
| **Mahalanobis** | 73.81 ± 3.71 | 73.52 ± 6.16 | 77.29 ± 3.18 | 72.39 ± 5.91 | 75.55±2.46 | 72.95 ± 0.80 |
| **NECO-Feature** | **76.20 ± 3.11** | **68.50 ± 6.01** | **77.98 ± 2.23** | 70.23 ± 4.23 | **77.09±1.26** | **69.36 ± 1.22** |
| **NECO** | 63.25 ± 2.34 | 88.62 ± 1.15 | 62.69 ± 3.02 | 89.81 ± 1.62 | 62.97±0.39 | 89.21 ± 0.85 |
| **NaN** | 61.26 ± 4.09 | 95.16 ± 2.23 | 62.88 ± 5.63 | 93.17 ± 4.50 | 62.07±1.15 | 94.16 ± 1.41 |
| **Residual** | 75.28 ± 3.85 | 71.65 ± 7.00 | 77.68 ± 3.24 | 71.76 ± 6.37 | 76.48±1.70 | 71.71 ± 0.08 |
| **ViM** | 71.46 ± 2.56 | 76.33 ± 4.65 | 72.53 ± 2.36 | 77.42 ± 4.17 | 71.99±0.75 | 76.87 ± 0.78 |
| **VECO-P (Ours)** | 75.92 ± 3.24 | 69.01 ± 6.22 | 77.90 ± 2.37 | **70.22 ± 4.41** | 76.91±1.40 | 69.61 ± 0.86 |
| **VECO-M (Ours)** | 73.87 ± 3.72 | 73.40 ± 6.15 | 77.33 ± 3.17 | 72.26 ± 5.87 | 75.60±2.45 | 72.83 ± 0.81 |

**Feature-based scores outperform logit and hybrid baselines.**   Across all multimodal settings, purely feature-based scores (Mahalanobis, Residual norm, and NECO-Feature) consistently outperform logit-based and hybrid alternatives. Hybrid scores that combine logits and features (e.g., NECO and ViM) often underperform relative to their own feature-only components. This suggests that the dominant OOD signal arises from feature-space rather than classifier confidence and that standard logit-feature fusion weakens the underlying OOD signal in these settings.

Notably, NaN, which operates directly on the full feature space via a global activation norm, consistently underperforms geometry-aware feature scores in these settings, often ranking closer to logit-based confidence measures. This indicates that utilizing feature representations alone is insufficient for robust OOD detection without explicitly exploiting the underlying geometric structure.

**VECO improves on strong feature baselines.**   Both VECO variants are competitive or superior to the tested baselines, including the best feature-based detectors. VECO-P achieves the best overall performance on both Tobacco3482 settings and attains the lowest average FPR95 on FinanceDocs, including in near-saturated regimes where AUROC is already close to 99%. VECO-M remains close to the top-performing feature baselines, but is consistently behind VECO-P on average in this modality.

Both VECO instantiations achieve strong performance on the document benchmarks. VECO-P consistently attains the best average performance across both datasets and OOD settings, while VECO-M performs competitively but slightly below. This gap suggests that ID conformity to a low-dimensional principal subspace is more aligned with this setting, and captures the most informative ID structure.

## 5.2   Text Classification (BERT)

Tables 3 and 4 report near- and far-OOD detection results for a BERT SST-2 classifier.

**Near-OOD: feature-based methods are strongest.**   Across near-OOD datasets (IMDB and Yelp), feature-based methods again dominate. NECO-Feature achieves the strongest overall performance, with Residual, Mahalanobis, and both VECO instantiations following closely. VECO-P performs on par with NECO-Feature in AUROC and yields comparable FPR95, while the Mahalanobis baseline closely trails VECO-M. In contrast, NaN provides no consistent improvement over logit-based baselines in this regime,

Table 4: **Far-OOD detection on SST-2.** ID = SST-2; OOD datasets = RTE, AG NEWS, and TREC-QC. AUROCand FPR95are reported (mean ± std over 10 seeds).

| Method | Far OOD | | | | | | | |
| | RTE | | AG News | | TREC-QC | | Average | |
| | AUROC | FPR95 | AUROC | FPR95 | AUROC | FPR95 | AUROC | FPR95 |
|---|---|---|---|---|---|---|---|---|
| **MSP** | 87.97 ± 1.39 | 68.35 ± 5.22 | 85.30 ± 1.46 | 74.80 ± 3.86 | 84.68 ± 3.54 | 84.54 ± 3.63 | 85.99±1.75 | 75.90 ± 8.15 |
| **MaxLogit** | 87.70 ± 2.22 | 64.51 ± 6.54 | 84.96 ± 2.22 | 71.71 ± 4.87 | 84.38 ± 3.53 | 83.21 ± 7.31 | 85.68±1.78 | 73.14 ± 9.43 |
| **KL-Matching** | 70.99 ± 4.97 | 68.27 ± 5.16 | 62.80 ± 4.17 | 74.56 ± 4.08 | 52.75 ± 6.27 | 85.07 ± 3.56 | 62.18±9.13 | 75.97 ± 8.49 |
| **GradNorm** | 56.94 ± 11.72 | 90.21 ± 5.76 | 56.18 ± 11.33 | 90.57 ± 5.11 | 60.92 ± 15.87 | 89.62 ± 8.32 | 58.01±2.55 | 90.13 ± 0.48 |
| **Energy** | 87.85 ± 2.34 | 62.71 ± 9.13 | 85.08 ± 2.30 | 70.35 ± 6.28 | 84.33 ± 3.79 | 83.92 ± 9.47 | 85.75±1.86 | 72.33 ± 10.74 |
| **Energy+React** | 54.24 ± 12.00 | 88.61 ± 14.28 | 53.33 ± 10.96 | 89.79 ± 11.92 | 56.11 ± 14.88 | 92.55 ± 10.41 | 54.56±1.42 | 90.32 ± 2.02 |
| **ASH-B** | 52.72 ± 22.39 | 93.49 ± 5.42 | 54.50 ± 23.88 | 91.30 ± 7.10 | 59.54 ± 27.61 | 89.00 ± 11.33 | 55.59±3.54 | 91.26 ± 2.24 |
| **ASH-P** | 51.05 ± 11.13 | 92.93 ± 11.85 | 50.86 ± 10.32 | 92.74 ± 8.30 | 54.22 ± 13.90 | 94.63 ± 7.85 | 52.05±1.89 | 93.43 ± 1.04 |
| **Mahalanobis** | 98.30 ± 0.48 | 8.87 ± 2.87 | 96.95 ± 0.54 | 15.97 ± 2.95 | 96.01 ± 1.16 | 21.82 ± 7.64 | 97.09±1.15 | 15.56 ± 6.49 |
| **NECO-Features** | 98.01 ± 0.70 | 11.95 ± 3.33 | 96.92 ± 0.64 | 16.13 ± 3.50 | 93.51 ± 1.74 | 36.46 ± 8.70 | 96.15±2.34 | 21.51 ± 13.11 |
| **NECO** | 87.92 ± 2.17 | 63.83 ± 6.75 | 85.22 ± 2.17 | 71.45 ± 4.89 | 84.51 ± 3.50 | 83.17 ± 7.33 | 85.88±1.80 | 72.82 ± 9.74 |
| **NaN** | 77.77 ± 6.19 | 83.77 ± 11.51 | 77.83 ± 5.61 | 82.83 ± 10.31 | 75.66 ± 10.18 | 89.64 ± 8.24 | 77.09±1.24 | 85.42 ± 3.69 |
| **Residual** | 98.29 ± 0.48 | 9.55 ± 2.90 | 96.99 ± 0.52 | 16.00 ± 2.97 | 95.85 ± 1.12 | 23.04 ± 7.16 | 97.04±1.22 | 16.20 ± 6.75 |
| **ViM** | 97.09 ± 0.62 | 15.87 ± 2.75 | 95.33 ± 0.72 | 24.27 ± 3.69 | 93.85 ± 1.53 | 37.21 ± 9.12 | 95.42±1.62 | 25.79 ± 10.75 |
| **VECO-P (Ours)** | 98.20 ± 0.61 | 10.11 ± 2.80 | **97.01 ± 0.59** | **15.33 ± 2.98** | 94.52 ± 1.47 | 29.98 ± 6.93 | 96.58±1.88 | 18.48 ± 10.30 |
| **VECO-M (Ours)** | **98.31 ± 0.47** | **8.79 ± 2.84** | 96.97 ± 0.54 | 15.85 ± 2.89 | **96.03 ± 1.16** | **21.67 ± 7.51** | **97.10±1.15** | **15.44 ± 6.45** |

despite operating in feature space. Hybrid methods (ViM and NECO) remain competitive but are consistently weaker than their feature-only counterparts.

**Far-OOD: VECO-M is best overall.** Under semantic and task shift, the strongest baselines are also feature-based (Mahalanobis and Residual). VECO-M achieves the best average performance and slightly improves on Mahalanobis across the far-OOD cases, while VECO-P remains competitive but is weaker on average in this regime. This suggests that for SST-2 features, which are highly anisotropic and arise from a binary classification task, a global covariance-based conformity signal more reliable than a purely principal-norm-based one.

**Summary.** Across both document and text classification benchmarks, the results reveal consistent trends: (i) feature-based OOD scores outperform logit-based and hybrid methods, (ii) standard hybridization often weakens rather than strengthens OOD signals, and (iii) the most effective VECO instantiation depends on the underlying representation geometry. VECO-P is best aligned with low-rank, class-rich document representations, while VECO-M is better suited to highly anisotropic text embeddings under semantic shift. Together, these results show that a shared soft-contrast principle, when paired with modality-aligned conformity signals, yields robust post-hoc OOD detection across modalities.

## 6 Conclusion

This work studied post-hoc out-of-distribution detection for textual and multimodal Transformer models, focusing on document classification with LayoutLMv3 and text classification with BERT. Across both modalities, we showed that unlike in many vision benchmarks, the dominant OOD signal resides in feature-space geometry rather than in classifier logits. Purely feature-based scores consistently outperform logit-based and hybrid methods, and standard logit–feature hybridization often degrade performance in these settings.

Building on this observation, we introduced **VECO**, a geometry-inspired, feature-space OOD scoring framework that implements a stable soft contrast between in-distribution conformity and residual-space deviation. We provided a likelihood-ratio motivation under a simple block-structured Gaussian model, offering intuition for why this contrast captures OOD-relevant structure in anisotropic embedding spaces. We instantiated VECO using principal-subspace conformity (VECO-P) for multimodal document models and Mahalanobis conformity (VECO-M) for text classifiers, reflecting modality-aligned representation geometry.

Empirically, VECO achieves state-of-the-art or consistently improved performance across multimodal document and text classification semantic-shift benchmarks. The results further show that the relative effectiveness of conformity signals is modality dependent: low-rank principal conformity is most informative

for document representations, while covariance-based conformity is more reliable for highly anisotropic text embeddings under semantic shift.

Overall, this work highlights that post-hoc OOD detection should be approached as a *modality-aware* problem. By explicitly accounting for the geometric structure of learned representations, simple geometry-aware feature-space scoring functions can yield stable and effective OOD detection without retraining or logit-based hybridization.

## 7 Broader Impact

Out-of-distribution (OOD) detection aims to improve the reliability of machine learning systems by identifying inputs that deviate from the training distribution. In NLP and multimodal document understanding, this can help reduce overconfident predictions on unfamiliar inputs and support safer deployment through abstention or human oversight.

However, OOD detection does not eliminate risks. Errors in OOD detection may still lead to incorrect or rejected predictions, and the method inherits biases present in the underlying data and representations. In particular, inputs from underrepresented groups or domains may be more likely to be flagged as out-of-distribution, raising potential fairness concerns. Additionally, OOD signals should not be used as definitive decisions, but rather as indicators of uncertainty within a broader system.

Overall, this work focuses on improving post-hoc reliability, and future work should further investigate fairness, robustness across diverse data, and integration into real-world decision-making pipelines.

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

# A   Details on Baselines Implementation

In this section, we provide additional implementation details for all baseline methods used in our experiments. For each method, we briefly describe the underlying scoring mechanism, the representation on which it operates (logits, features, or both), and any hyperparameters required for fair comparison.

**Maximum Softmax Probability (MSP).**   The MSP score (Hendrycks & Gimpel, 2017) uses the maximum softmax probability of the classifier as an OOD score. Given logits $z(x)$, the score is defined as

$$\text{MSP}(x) = \max_c \sigma_c(z(x)),$$

where $\sigma$ denotes the softmax function. Lower maximum probability values indicate higher likelihood of being OOD.

**MaxLogit.**   MaxLogit (Hendrycks et al., 2022) uses the maximum pre-softmax logit:

$$\text{MaxLogit}(x) = \max_c z_c(x).$$

Unlike MSP, this score avoids the normalization induced by the softmax transformation and has been shown to improve robustness in some settings.

**Energy.**   The Energy score (Liu et al., 2020a) maps logits to a scalar using

$$E(x) = -\log \sum_c e^{z_c(x)}.$$

Following standard convention, lower (more negative) energy values correspond to in-distribution samples. Energy-based scoring can be interpreted as a softened alternative to MaxLogit.

**KL-Matching.**   KL-Matching (Hendrycks et al., 2022) compares the predictive distribution of a test sample to class-wise average predictive distributions computed over the training set. The class averages are estimated using predicted labels, and the OOD score is defined as the minimum KL divergence to these reference distributions.

**GradNorm.**   (Huang et al., 2021) computes the norm of gradients between the softmax output and a uniform probability distribution.

**Mahalanobis.**   The Mahalanobis score (Lee et al., 2018) computes class-conditional Gaussian statistics in feature space. Let $f(x)$ denote the representation extracted before the final classification layer. The covariance matrix $\Sigma$ is calculated over the full in-distribution training set, whereas the empirical mean $\mu_c$ is estimated for each class $c$. The score is defined as

$$\text{Mahalanobis}(x) = -\min_c (f(x) - \mu_c)^\top \Sigma^{-1} (f(x) - \mu_c).$$

Ground-truth labels are used only for estimating the class statistics from training data. At test time, no label information is used.

**Residual.**   The Residual score (Wang et al., 2022) corresponds to the norm of the feature projection onto the orthogonal complement of the principal subspace:

$$\text{Residual}(x) = \|P^\perp f(x)\|_2.$$

This score isolates deviation from the dominant subspace without incorporating logit information.

**ReAct + Energy.** ReAct (Sun et al., 2021) performs activation clipping in feature space prior to computing an OOD score. Given a clipping threshold determined from the in-distribution training activations (typically a high percentile), feature activations exceeding this threshold are truncated. We combine ReAct with the Energy score, as this configuration was reported to be effective in prior work. The clipping percentile is selected based on validation performance on held-out in-distribution data.

**ViM.** ViM (Wang et al., 2022) decomposes the feature space into a principal subspace and its orthogonal complement. Let $P$ denote the principal subspace estimated via PCA on in-distribution features, and $P^\perp$ its orthogonal complement. The norm of the projection onto $P^\perp$ is used to construct a "virtual logit," which is combined with the original logits to produce the final score. A scaling factor $\alpha$ is computed on the training set to calibrate the contribution of the residual norm relative to the logits.

**NECO.** *NECO* (Ammar et al., 2024) introduced the use of the simplex ETF subspace for OOD detection. Its score measures the ratio between a feature vector's projection onto this subspace and the full vector norm, quantifying alignment with the collapsed representations manifold. This norm is then multiplied by the maximum logit output.

**ASH.** ASH, as introduced by Djurisic et al. (2022), employs activation pruning at the penultimate layer, just before the application of the DNN classifier. This pruning threshold is determined on a per-sample basis, eliminating the need for pre-computation of ID data statistics. The original paper presents three different post-hoc scoring functions, with the only distinction among them being the imputation method applied after pruning. In ASH-P, the clipped values are replaced with zeros. ASH-B substitutes them with a constant equal to the sum of the feature vector before pruning, divided by the number of kept features. ASH-S imputes the pruned values by taking the exponential of the division between the sum of the feature vector before pruning and the feature vector after pruning. We have used a threshold of 75%, as it provided the best results.

## B  Derivation of the VECO Score

We provide additional details for the log-likelihood ratio that motivates the VECO score introduced in Section 3.3. The model is not intended as a faithful density model, but as a minimal abstraction that isolates the dominant geometric contrast observed empirically, we do not claim the feature distribution is Gaussian nor that ID/OOD share eigenvectors.

Throughout Appendix B, we adopt an idealized setting that enables closed-form analysis. In particular, we assume: (i) the feature space admits a decomposition into principal and residual subspaces; (ii) the corresponding covariance matrices are simultaneously diagonalizable in the chosen basis; and (iii) principal and residual components are conditionally independent under both ID and OOD distributions. These assumptions are not meant to hold exactly in practice, but allow us to isolate and analyze the geometric role of principal conformity versus residual deviation in OOD detection.

### B.1  Generative Model

We consider a class-marginal Gaussian abstraction expressed in the PCA basis of in-distribution (ID) features. we assume the leading PCA directions align with the class-mean subspace, as suggested by NC-like geometry. Let $z_p \in \mathbb{R}^k$ and $z_r \in \mathbb{R}^m$ denote the principal and residual coordinates of a centered feature vector, respectively.

For in-distribution features, we assume

$$z_p \sim \mathcal{N}(0, \Sigma_p^{\mathrm{ID}}), \qquad \Sigma_p^{\mathrm{ID}} = \sigma_p^2 I_k + \Sigma_{\mathrm{disc}},$$

Since we express $z_p = U_k^\top f$ in a basis that we assume aligns with the (class-mean) discriminative subspace, we adopt an idealized assumption that the class-discriminative covariance $\Sigma_{\mathrm{disc}}$ is diagonal in this basis.

Concretely, we assume the first $C$ coordinates capture the dominant class-mean structure (with $C$ the number of ID classes). For clarity, we set $k = C$ to match the number of in-distribution classes, with

$$\Sigma_{\text{disc}} = \text{diag}(\lambda_1, \ldots, \lambda_C), \qquad \lambda_i > 0,$$

(and $\Sigma_{\text{disc}}$ is zero outside this $C$-dimensional subspace when embedded in $\mathbb{R}^k$).

Residual coordinates are modeled as

$$z_r \sim \mathcal{N}(0, \sigma_r^2 I_m).$$

For out-of-distribution (OOD) features, we assume isotropic covariance in both blocks:

$$z_p \sim \mathcal{N}(0, \tilde{\sigma}_p^2 I_k), \qquad z_r \sim \mathcal{N}(0, \tilde{\sigma}_r^2 I_m).$$

Empirically, we typically observe that $\sigma_p^2$ and $\tilde{\sigma}_p^2$ have comparable scale, while $\sigma_r^2 < \tilde{\sigma}_r^2$, reflecting stronger dispersion of OOD samples in residual directions.

For clarity, the covariance matrices are defined as:

$$\Sigma_p^{\text{ID}} = \sigma_p^2 I_k + \Sigma_{\text{disc}}, \qquad \Sigma_p^{\text{OOD}} = \tilde{\sigma}_p^2 I_k,$$

$$\Sigma_r^{\text{ID}} = \sigma_r^2 I_m, \qquad \Sigma_r^{\text{OOD}} = \tilde{\sigma}_r^2 I_m.$$

**NC/ETF alignment (idealization).** Under Neural Collapse–like geometry, class means concentrate in a $(C-1)$-dimensional simplex ETF subspace, and the between-class (class-mean) scatter has (approximately) equal nonzero eigenvalues on that subspace. Motivated by this, we idealize that the chosen principal basis $U_k$ aligns with the class-mean subspace (equivalently, one may take $U_k$ as the eigenbasis of the empirical between-class scatter matrix), so that the discriminative component is diagonal in this basis and its diagonal entries $\lambda_i$ have comparable scale for balanced classes. Note that the models used in our empirical results section all exhibit NC-like geometry, as shown in Figures 8 and 2.

## B.2 Log-likelihood Ratio

We derive the log-likelihood ratio induced by the Gaussian abstraction introduced above and show how it decomposes into principal- and residual-space contributions.

For a zero-mean Gaussian distribution with covariance matrix $\Sigma$, the log-density is

$$\log p(z) = -\frac{1}{2} z^\top \Sigma^{-1} z - \frac{1}{2} \log \det \Sigma + \text{const.}$$

Therefore, for the principal coordinates $z_p$, the log-likelihood ratio is

$$\log \frac{p(z_p \mid \text{ID})}{p(z_p \mid \text{OOD})} = -\frac{1}{2} z_p^\top \left[ (\Sigma_p^{\text{ID}})^{-1} - (\Sigma_p^{\text{OOD}})^{-1} \right] z_p + \frac{1}{2} \left( \log \det \Sigma_p^{\text{OOD}} - \log \det \Sigma_p^{\text{ID}} \right)$$

$$\propto \frac{1}{2} z_p^\top \left[ (\Sigma_p^{\text{OOD}})^{-1} - (\Sigma_p^{\text{ID}})^{-1} \right] z_p.$$

Since determinant terms do not depend on the sample $z$, they do not affect ranking and are omitted in what follows.

**Block decomposition.** By construction, the principal and residual coordinates $(z_p, z_r)$ are independent under both the ID and OOD models. Consequently, the log-likelihood ratio decomposes additively:

$$\log \frac{p(f \mid \text{ID})}{p(f \mid \text{OOD})} = \log \frac{p(z_p \mid \text{ID})}{p(z_p \mid \text{OOD})} + \log \frac{p(z_r \mid \text{ID})}{p(z_r \mid \text{OOD})}.$$

We analyze each term separately.

**Principal block.** Recall that we set $k = C$ in this appendix, the principal-space covariances are,

$$\Sigma_p^{\text{ID}} = \sigma_p^2 I_k + \Sigma_{\text{disc}} = \text{diag}(\sigma_p^2 + \lambda_1, \ldots, \sigma_p^2 + \lambda_C), \qquad \Sigma_p^{\text{OOD}} = \tilde{\sigma}_p^2 I_k.$$

Both matrices are diagonal in this PCA-aligned idealization, and therefore commute and share eigenvectors, we obtain

$$\left(\Sigma_p^{\text{OOD}}\right)^{-1} - \left(\Sigma_p^{\text{ID}}\right)^{-1} = \text{diag}\left(\frac{1}{\tilde{\sigma}_p^2} - \frac{1}{\sigma_p^2 + \lambda_1}, \ldots, \frac{1}{\tilde{\sigma}_p^2} - \frac{1}{\sigma_p^2 + \lambda_C}\right).$$

Define the weights

$$w_i = \frac{1}{\tilde{\sigma}_p^2} - \frac{1}{\sigma_p^2 + \lambda_i} > 0 \quad (\text{since } \tilde{\sigma}_p^2 < \lambda_i + \sigma_p^2)$$

and let $W = \text{diag}(w_1, \ldots, w_C)$. Then the principal term can be written compactly as

$$\log \frac{p(z_p \mid \text{ID})}{p(z_p \mid \text{OOD})} \propto \frac{1}{2} z_p^\top W z_p = \frac{1}{2} \|W^{1/2} z_p\|_2^2,$$

where $W^{1/2} = \text{diag}(\sqrt{w_1}, \ldots, \sqrt{w_C})$. Note that $w_i$ increases with larger $\lambda_i$, highlighting the utility of this geometric concentration.

**Residual block.** Similarly, with $\Sigma_r^{\text{ID}} = \sigma_r^2 I_m$ and $\Sigma_r^{\text{OOD}} = \tilde{\sigma}_r^2 I_m$,

$$\log \frac{p(z_r \mid \text{ID})}{p(z_r \mid \text{OOD})} \propto \frac{1}{2} z_r^\top \left[\left(\Sigma_r^{\text{OOD}}\right)^{-1} - \left(\Sigma_r^{\text{ID}}\right)^{-1}\right] z_r = -\frac{1}{2} \gamma_r \|z_r\|_2^2,$$

where

$$\gamma_r = \frac{1}{\sigma_r^2} - \frac{1}{\tilde{\sigma}_r^2} > 0 \quad (\text{since } \sigma_r^2 < \tilde{\sigma}_r^2).$$

**Complete form.** Combining the independent blocks, we obtain (up to an additive constant independent of $(z_p, z_r)$)

$$\Lambda(f) = \log \frac{p(f \mid \text{ID})}{p(f \mid \text{OOD})} \propto \frac{1}{2} \|W^{1/2} z_p\|_2^2 - \frac{1}{2} \gamma_r \|z_r\|_2^2.$$

This expression makes clear that the likelihood ratio favors small residual energy $\|z_r\|_2^2$, and favors concentration along the principal space.

To connect more directly to the two-scalar VECO template, we consider the balanced-class simplification $\lambda_1 \approx \cdots \approx \lambda_C \approx \bar{\lambda}$, which yields $w_i \approx \bar{w}$ and hence

$$\|W^{1/2} z_p\|_2^2 \approx \bar{w} \|z_p\|_2^2 = \bar{w} s_p^2, \qquad s_p = \|z_p\|_2, \ \ s_r = \|z_r\|_2.$$

Under this approximation,

$$\boxed{\Lambda(f) \propto \frac{1}{2} \bar{w} s_p - \frac{1}{2} \gamma_r s_r, \qquad \bar{w}, \gamma_r > 0.}$$

Importantly, while the likelihood ratio and the VECO score are not required to share identical coefficients or induce identical rankings, they depend on the same two scalar statistics,

$$s_p = \|z_p\|_2^2 \quad \text{and} \quad s_r = \|z_r\|_2^2,$$

and favor them in the same qualitative directions. In particular, both the idealized likelihood ratio and the VECO score increase monotonically with increasing principal energy and decrease monotonically with increasing residual energy. The VECO formulation can therefore be viewed as a smooth, contrastive surrogate that captures the same principal–residual trade-off highlighted by the likelihood analysis, without attempting to replicate its exact functional form or parameterization.

We emphasize that the likelihood-based analysis in this appendix is intended purely as a motivational abstraction. The VECO score is not derived as a plug-in approximation of the likelihood ratio, nor does it assume a Gaussian feature distribution in practice. Instead, the analysis serves to highlight a geometric principle: effective OOD detection can be achieved by contrasting conformity within a principal (task-aligned) subspace against deviation in its residual complement. The specific definitions of the VECO components are chosen to (i) align with this principle, (ii) remain numerically stable and scalable in high dimensions, and (iii) connect naturally to existing hybrid OOD scores such as ViM and NECO that utilize feature norms, while unifying them under a common feature-based principal–residual contrast framework.

**Choice of $k$.** The rank $k$ controls how much of the ID representation manifold is treated as principal structure. For LayoutLMv3 document classifiers, the eigenvalue spectra in Figure 1 exhibit a sharp spectral drop followed by a long flat tail. This pattern is consistent with documented anisotropy in contextual representations (Ethayarajh, 2019; Gao et al., 2021) and with collapse-like behavior observed in classification models (Papyan et al., 2020). These geometric properties further motivate the principal–residual subspace decomposition in VECO. In this setting, we find that a very low-rank principal space is sufficient; in particular, $k=1$ yields stable behavior across all document benchmarks, as the top PCs captures global structure shared by all classes and not class-means aligned directions. Larger $k$ values led to similar but slightly weaker performance, (e.g. using $k = 256$ changes AUROC/FPR95 marginally on FinanceDocs, having no effect on score ordering). For BERT on SST-2, where a single component can dominate variance (see Figure 10), we set $k$ to the smallest rank capturing 99% of the variance, matching the NECO's (Ammar et al., 2024) hyperparameter setting. This ensures $s_p$ reflecting broader ID structure rather than only the dominant direction signal. These choices reflect the differences between the two modalities and tasks.

### B.3 Equivalence of VECO Score Forms

Starting from equation 2,

$$
\begin{aligned}
S_{\text{VECO}}(\mathbf{f}) &= s_p - \log\big(e^{s_p} + e^{\alpha s_r}\big) \\
&= s_p - \log\Big(e^{s_p}\big(1 + e^{\alpha s_r - s_p}\big)\Big) \\
&= s_p - \Big(s_p + \log\big(1 + e^{\alpha s_r - s_p}\big)\Big) \\
&= -\log\big(1 + e^{\alpha s_r - s_p}\big),
\end{aligned}
$$

which yields equation 6.

## C Hyperparameter sensitivity analysis

This section presents ablation results on the principal subspace dimension $k$, residual subspace dimension $m$, and the use of the calibration factor $\alpha$. Overall, the results show that VECO is robust to the choice of hyperparameters and does not require precise tuning. Detailed numerical results and sensitivity plots are reported in Table 5 and Figures 4, 5, 6, 7.

These observations are consistent with the geometric intuition behind VECO: once the principal subspace captures the dominant representation directions, the contrast between principal conformity and residual deviation becomes stable, making the score largely insensitive to moderate variations of $k$ and $m$.

## D Qualitative Analysis

To complement quantitative metrics, we report representative *extreme* examples from the VECO-P score distribution: (i) **ID false positives** (ID samples flagged as OOD), and (ii) **OOD false negatives** (OOD samples not flagged as OOD). These examples illustrate typical failure patterns such as short/ambiguous ID inputs and OOD samples with stylistic similarity to ID. **OOD decision rule.** We flag a sample as OOD when $\text{VECO}(x) < \tau$, where $\tau$ is set on ID validation to achieve 95% TPR (VECO-P$(x) < -24.31$,

Table 5: Ablation on the calibration factor $\alpha$. We report the **average FPR95** corresponding to the last column of multimodal benchmarks (LayoutLMv3), and text benchmarks (BERT). The results show that $\alpha$ consistently improves or stabilizes performance by calibrating the relative magnitudes of principal and residual energies.

| | VECO-P | | VECO-M | |
| --- | --- | --- | --- | --- |
| Benchmark | with $\alpha$ | w/o $\alpha$ | with $\alpha$ | w/o $\alpha$ |
| Multimodal: Tobacco3482 & Tobacco3482 (ADVE) | **43.90 ± 5.20** | 48.01 ± 7.01 | **49.78 ± 5.55** | 49.80 ± 5.71 |
| Multimodal: FinanceDocs & FinanceDocs (Resumes) | **3.04 ± 1.23** | 5.46 ± 3.96 | **3.35 ± 3.87** | 3.36 ± 3.87 |
| Text: Far OOD | **18.48 ± 10.30** | 20.04 ± 7.53 | 15.44 ± 6.45 | **15.43 ± 6.44** |
| Text: Near OOD | **69.61 ± 0.86** | 77.50 ± 2.48 | **72.83 ± 0.81** | 72.88 ± 0.83 |

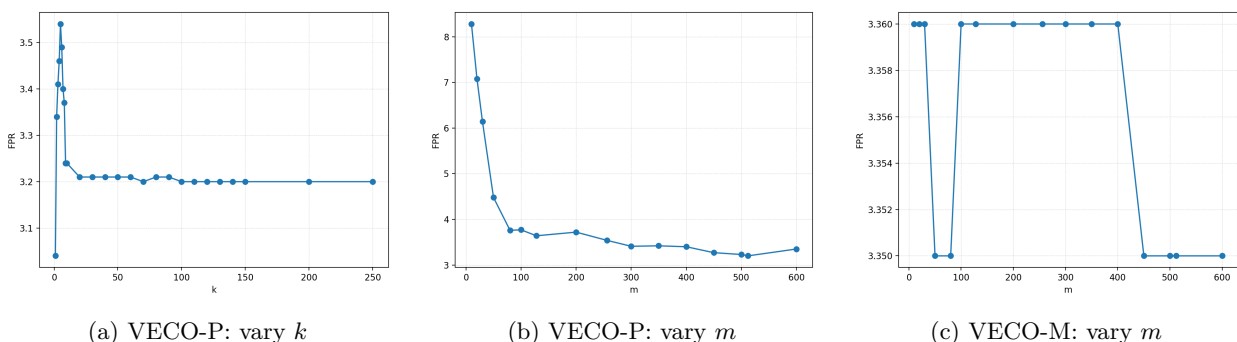

(a) VECO-P: vary $k$       (b) VECO-P: vary $m$       (c) VECO-M: vary $m$

Figure 4: Hyperparameter sensitivity analysis on the **FinanceDocs** benchmarks. Each panel reports the **average FPR95** when varying the principal dimension $k$ and the residual dimension $m$. Performance remains stable for a wide range of hyperparameters, indicating that VECO does not require precise tuning in this setting.

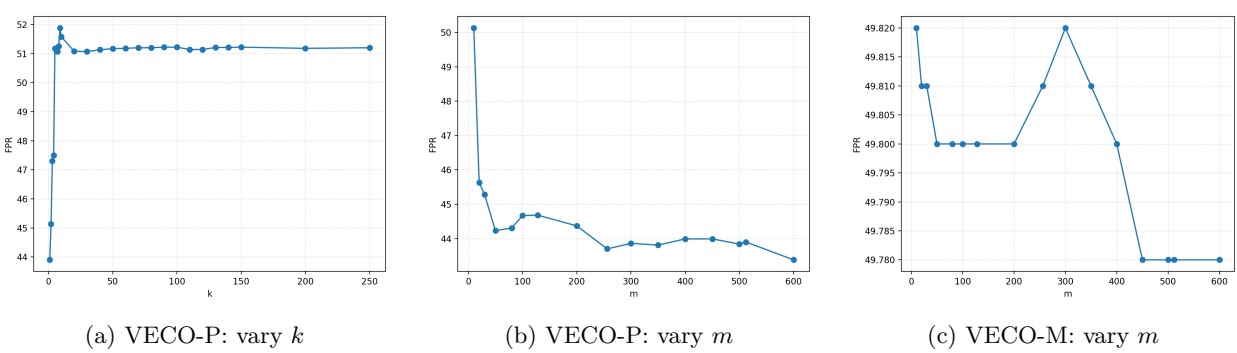

(a) VECO-P: vary $k$       (b) VECO-P: vary $m$       (c) VECO-M: vary $m$

Figure 5: Hyperparameter sensitivity analysis on the **Tobacco3482** benchmarks. Each panel reports the **average FPR95** when varying the principal dimension $k$ and the residual dimension $m$. Results show that VECO is relatively stable across a broad range of values for $m$, while being more sensitive to the $k$ value, which must capture only the dominant ID feature geometry.

VECO-M$(x) < -2377.86$). VECO-P achieves AUROC 98.20 / FPR@95 10.11 on RTE, while TREC-QC is more challenging (AUROC 94.52 / FPR@95 29.98), which is reflected in the qualitative failure cases below.

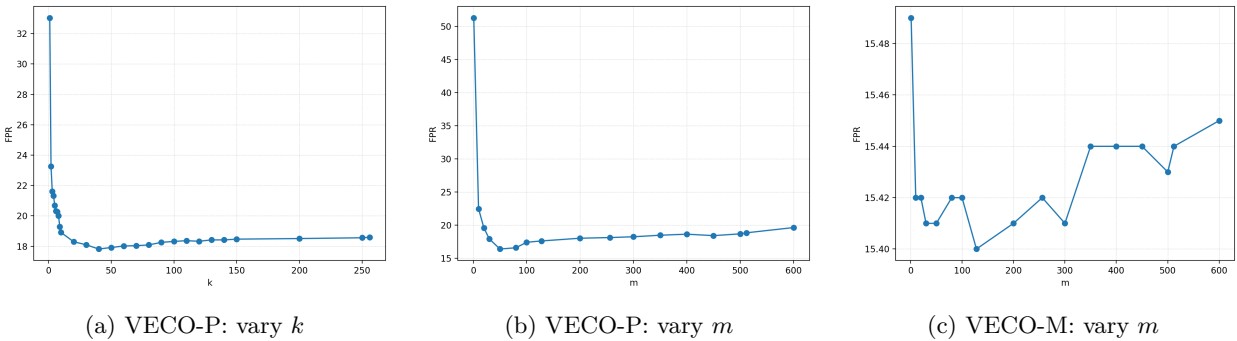

(a) VECO-P: vary $k$       (b) VECO-P: vary $m$       (c) VECO-M: vary $m$

Figure 6: Hyperparameter sensitivity analysis on the **Far-OOD text** benchmarks. **Average FPR95** as he principal dimension $k$ and residual dimension $m$ are varied. Performance remains stable for a wide range of hyperparameters, indicating that VECO does not require precise tuning in this setting.

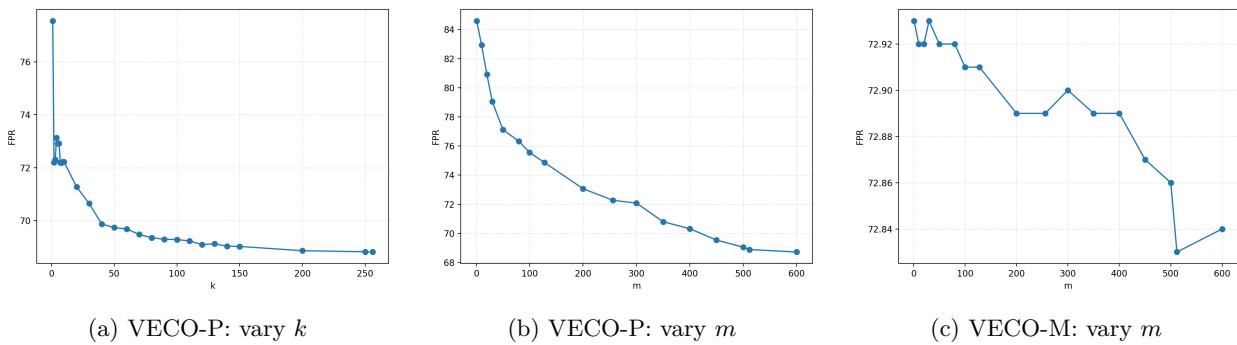

(a) VECO-P: vary $k$       (b) VECO-P: vary $m$       (c) VECO-M: vary $m$

Figure 7: Hyperparameter sensitivity analysis on the **Near-OOD text** benchmarks. **Average FPR95** as he principal dimension $k$ and residual dimension $m$ are varied. The method shows robustness to hyperparameter choices with continuous improvement when increasing $k$ or $m$, supporting the use of fixed large values across experiments.

Table 6: Representative **ID false positives** for VECO-P on SST-2 validation (flagged as OOD). Pred/Conf are the classifier prediction (task label), not the ID/OOD label, and its confidence in the prediction. *Pred* denotes the classifier prediction. OOD detection is determined by the VECO score and the threshold $\tau$.

| ID text (SST-2 test) | Pred | Conf | VECO-P | VECO-M | Flagged OOD |
|---|---|---|---|---|---|
| rarely has leukemia looked so shimmering and benign . | 0 | 0.657 | -40.017 | -2392.44 | Yes |
| oh come on . | 0 | 0.574 | -37.510 | -2389.95 | Yes |
| if you believe any of this , i can make you a real deal on leftover enron stock that will double in value a week from friday . | 0 | 0.874 | -36.453 | -2388.71 | Yes |

**Discussion.** Two common patterns appear in the qualitative examples. First, very short or highly sarcastic ID sentences can appear atypical and thus be flagged as OOD. Second, certain OOD inputs (e.g., factual questions or news-style sentences) may exhibit representation geometry similar to ID movie-review sentences, leading to high conformity scores and missed detections. Both VECO-P and VECO-M highlight largely identical qualitative failure cases, suggesting that these errors arise from the underlying representation geometry rather than the specific conformity signal used in the score.

Table 7: Representative **OOD false negatives** for VECO-P: RTE examples that are **not flagged** as OOD (highly ID-like by the score). For readability, long texts are clipped to the first N characters. Pred/Conf are the classifier prediction (task label), not the ID/OOD label, and its confidence in the prediction.

| OOD text (RTE, truncated) | Pred | Conf | VECO-P | VECO-M | Flagged OOD |
|---|---|---|---|---|---|
| We look at the cool relationship between these two establishment families and how the party would fare with the son, Texas Gov. George W. Bush, and the wife, Elizabeth Dole, on the ... | 1 | 0.998 | -8.279 | -2361.92 | No |
| The beleaguered Euro-Disney theme park outside Paris is doing so poorly it might have to close unless it gets help soon from its lenders, the chairman of Walt Disney Co. said in an ... | 0 | 0.997 | -9.136 | -2363.18 | No |
| Arlene Blum is a legendary trailblazer by any measure. Defying the climbing establishment of the 1970s, she led the first teams of women on successful ascents of Mt. McKinley and A ... | 1 | 0.999 | -12.679 | -2366.74 | No |

Table 8: Representative **OOD false negatives** for VECO-P: TREC-QC examples that are **not flagged** as OOD. Pred/Conf are the classifier prediction (task label), not the ID/OOD label, and its confidence in the prediction.

| OOD text (TREC-QC) | Pred | Conf | VECO-P | VECO-M | Flagged OOD |
|---|---|---|---|---|---|
| What is mold ? | 0 | 0.998 | -0.011 | -2352.04 | No |
| What is an obtuse angle ? | 0 | 0.997 | -0.012 | -2352.18 | No |
| What is done with worn or outdated flags ? | 0 | 0.998 | -0.016 | -2352.41 | No |

# E  Additional Geometry Diagnostics

This appendix provides additional empirical diagnostics supporting the geometric assumptions underlying VECO. We visualize class structure and OOD dispersion in both principal and residual PCA directions across multiple multimodal document benchmarks.

**Class structure in the principal subspace.**  Across all document settings, projections onto the leading principal components reveal compact, well-separated in-distribution class clusters, consistent with collapse-like geometry. OOD samples exhibit weak alignment with these discriminative directions and tend to concentrate near the origin of the principal subspace.

**Residual variance amplification for OOD samples.**  In contrast, projections onto high-index, low-variance PCA components show that in-distribution features are still tightly concentrated but collapse to the origin. In contrast, OOD samples exhibit substantially larger dispersion along residual directions, consistent with the variance assumptions used in our block-structured Gaussian abstraction.

**Score distributions.**  We further report distributions of the principal conformity score $s_p$, residual magnitude $s_r$, and VECO scores. While $s_p$ and $s_r$ alone yield only substantial separation, their soft contrast produces improved ID/OOD discrimination, explaining the empirical performance gains observed for VECO.

# F  BERT Representation Diagnostics

Figure 10 shows the eigenvalue spectrum of penultimate-layer features extracted from a BERT-base model fine-tuned on SST-2. The spectrum is highly anisotropic, with a single dominant component capturing a large fraction of the variance, followed by a slowly decaying tail. This structure differs markedly from typical

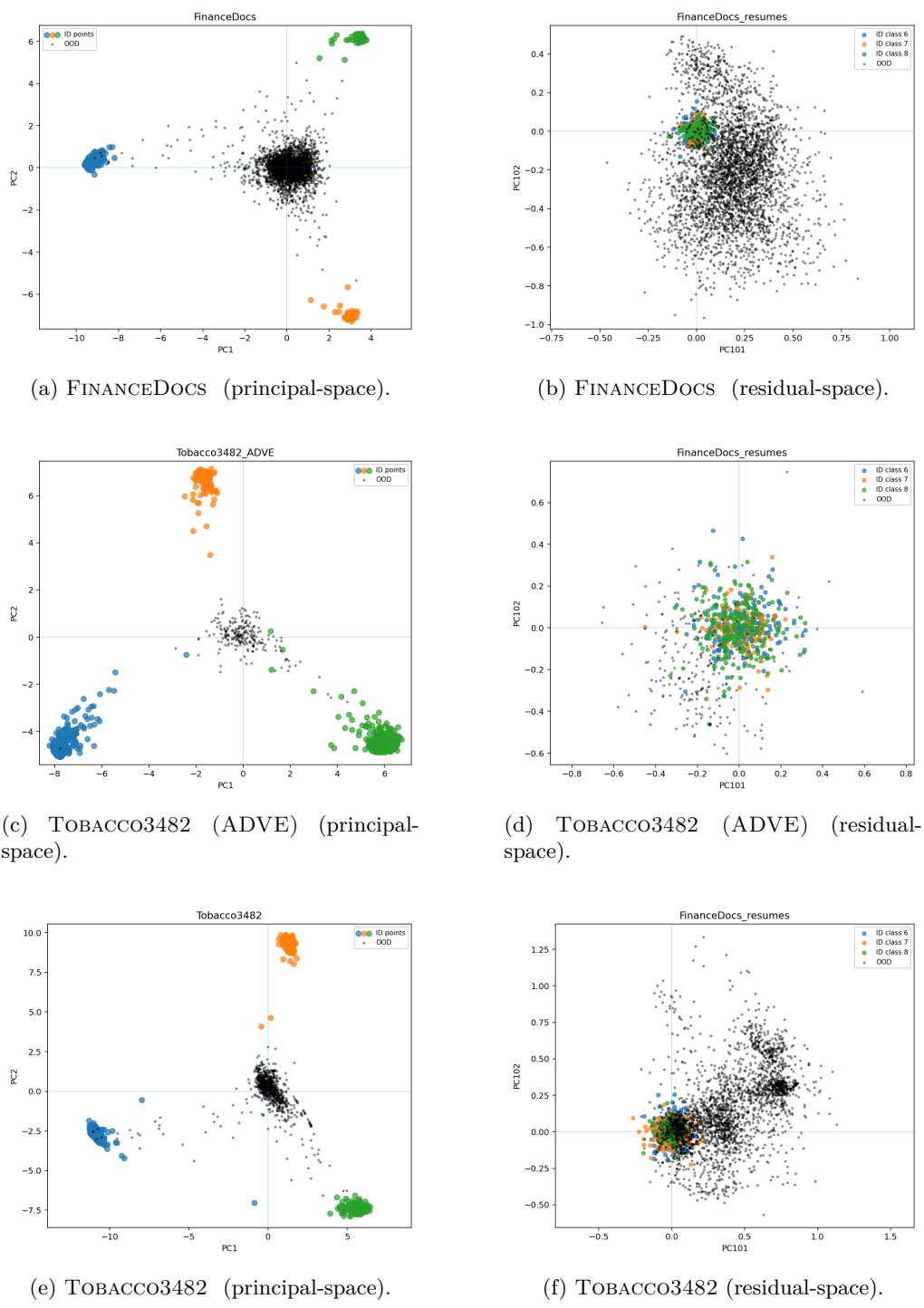

(a) FINANCEDOCS  (principal-space).

(b) FINANCEDOCS  (residual-space).

(c) TOBACCO3482 (ADVE) (principal-space).

(d) TOBACCO3482 (ADVE) (residual-space).

(e) TOBACCO3482 (principal-space).

(f) TOBACCO3482 (residual-space).

Figure 8: PCA projections of penultimate-layer features for the FinanceDocs (top), Tobacco3482 (ADVE) (middle), and Tobacco3482 (bottom) cases. (*Left*) Projection onto the top principal components (PC1–PC2).. (*Right*) Projection onto low-variance components (PC100–PC101). For clarity, we visualize three random ID classes (in colors), while OOD samples are shown in gray.

vision backbones, and empirically favors the use of covariance-aware conformity signals, such as Mahalanobis distance, in conjunction with residual deviations in the VECO score.

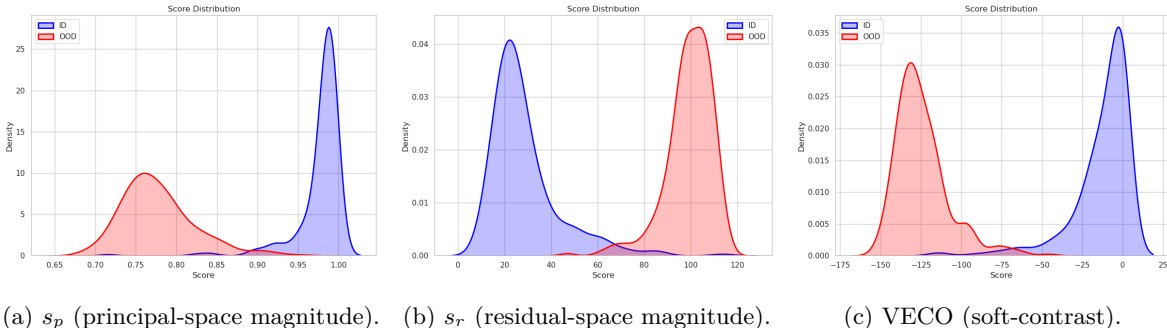

(a) $s_p$ (principal-space magnitude). (b) $s_r$ (residual-space magnitude). (c) VECO (soft-contrast).

Figure 9: Distributions of principal conformity ($s_p$), residual magnitude ($s_r$), and VECO scores for in-distribution (ID) and out-of-distribution (OOD) samples. While $s_p$ and $s_r$ individually provide partial separation, their soft contrast in VECO yields substantially improved discrimination, validating the likelihood-ratio–inspired formulation.

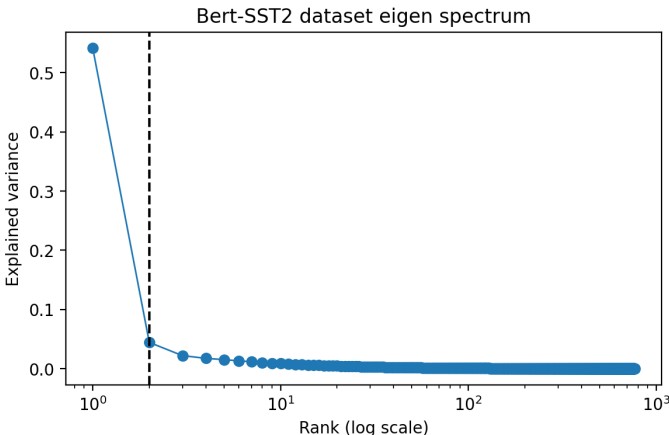

Figure 10: Eigenvalue spectrum of BERT-base penultimate-layer features on SST-2. The vertical dashed line marks the number of ID classes (2).

## G   Additional Discussion: OOD Difficulty in Document Benchmarks

A common intuition is that held-out-class OOD within a dataset should be "closer" to the in-distribution data than cross-dataset OOD, and therefore harder to detect. However, our document benchmarks do not consistently follow this ordering: held-out classes such as *Advertisement* (Tobacco3482) and *Resumes* (FinanceDocs) can yield AUROC and FPR95 comparable to, or even better than, cross-dataset OOD.

A plausible explanation is that certain document classes correspond to highly distinct layout and visual templates, inducing strong shifts in spatial structure and multimodal attention patterns despite originating from the same dataset. Such class-level shifts can therefore be as geometrically salient as broader domain-level changes. A more systematic geometric characterization of what determines the effective "distance" between document classes and datasets is left for future work.

## H   Algorithmic Details

Algorithm 1 summarizes the computation of the VECO score. All steps are performed in a post-hoc manner using only in-distribution training features, without retraining or access to OOD data outside of the evaluation step.

---

**Algorithm 1** VECO score (vector conformity)

---

**Require:** ID training features $\{\mathbf{f}_i^{\text{train}}\}$, test feature $\mathbf{f}$, PCA rank $k$, residual rank $r$
 1: Standardize all features.
 2: Fit PCA on ID training features and extract top-$k$ components $\mathbf{U}_k$.
 3: Extract $r$ lowest-variance components $\mathbf{U}_r$.
 4: Compute $s_p = \|\mathbf{U}_k^\top \mathbf{f}\|$ and $s_r = \|\mathbf{U}_r^\top \mathbf{f}\|$.
 5: Compute $\alpha$ using Eq. equation 3 on ID training features.
 6: **return** $S_{\text{VECO}}(\mathbf{f}) = s_p - \log\!\left(e^{s_p} + e^{\alpha s_r}\right)$.

---

