# OpenReview forum: "VECO: VEctor COnformity Based OOD Detection in Text and Multimodal Models"
_TMLR — Accepted by TMLR_

### Review · Reviewer_va86 · 2026-02-20

**Summary Of Contributions:**

This paper proposes a new out-of-distribution (OOD) detection method named VECO (VEctor COnformity). The main claim is that most existing approaches are post-hoc OOD detection methods that are designed for vision models and thus achieve suboptimal performance on text and multimodal models. To address this, the paper proposes a geometry-aware, purely feature-based OOD scoring framework, showing superior performance. The analysis also highlights the modality-dependent nature of OOD detection.

Strengths:
1. The paper is well written and easy to follow. The organization and narrative are good.
2. The motivation and the claim are clear. It first reveals that existing post-hoc OOD detection methods are designed for vision models and are thus suboptimal for text and multimodal models. Based on this observation, it proposes a feature-based framework and achieves improvement. The finding of the modality-dependent nature of OOD detection is also novel. Overall, this paper provides a new perspective and valuable insights for OOD detection across different modalities.
3. Section 4 provides detailed analysis based on toy verification, which supports the claim well. The analysis is interesting and technically sound.
4. The experimental results demonstrate that the proposed method achieves state-of-the-art performance.

Weaknesses:
1. The evaluation is conducted on only one model for each task (LayoutLMv3 for multimodal document classification and BERT for text classification), which is insufficient. More models should be applied to demonstrate the generalizability of the proposed method.
2. There are no qualitative results. The paper could present some examples of the text and document inputs along with the corresponding output predictions.
3. It is suggested to present and discuss some failure results.

**Audience:**

Yes

**Audience Explanation:**

Yes, audiences from the OOD domain would be interested in this work.

**Broader Impact Concerns:**

There is no Broader Impact Statement.

**Claims And Evidence:**

Yes

**Claims Explanation:**

Please see Strengths.

**Requested Changes:**

Please see Weaknesses.

---

> ### Author Response · Authors · 2026-03-09
>
> We thank the reviewer for the careful reading and encouraging assessment of our work. We appreciate the recognition of the clarity of the presentation, the modality-dependent perspective on OOD detection, and the strength of the experimental results.
>
> The reviewer raised several helpful suggestions regarding evaluation on additional models, qualitative examples, failure case analysis, and the inclusion of a broader impact statement. We address each of these points below. Corresponding **additions are highlighted in blue** in the revised manuscript.
>
> ##  Generalization of VECO
>
> We thank the reviewer for highlighting the importance of evaluating OOD detection methods across multiple architectures. In this work, we focused on two representative and widely used models: BERT for text classification and LayoutLMv3 for multimodal document understanding, similar to [3]. These models are commonly used benchmarks in their respective domains and provide a strong testbed for analyzing OOD signals in transformer-based representations.
>
> Importantly, VECO is architecture-agnostic: being feature-based and post-hoc, it does not depend on model-specific components, but on the **geometric structure of learned representations**. When representations exhibit dominant principal directions and residual dispersion, a behavior commonly observed in modern deep representations and related to Neural-Collapse-like geometry [1,2], we expect VECO’s contrast mechanism to remain applicable.
>
> Due to space and compute constraints we limited the experiments to these representative models in the current submission. **We will clarify this scope in the revised manuscript and broader impact statement.**
>
> ## Qualitative results & Failure cases
>
> The original manuscript included geometric visualizations (Figures 1–3) illustrating representation structure. In response to the reviewer’s suggestion, we additionally introduce explicit qualitative analysis in **Appendix D**. We present representative ID and OOD examples ranked by their VECO scores, including both successful detections and failure cases. These examples highlight typical patterns such as short or stylistically atypical ID inputs and OOD samples that share similar representation geometry with ID data.
>
> These observations complement the geometric and toy verification analyses in Section 4, providing additional intuition about the behavior of the proposed score.
>
> As an interpretive lens for expected failure patterns, VECO may be less effective when representations become close to isotropic, or when principal/residual separation is weak, or when ID and OOD samples share similar geometric structure.
>
> ## Broader Impact Statement
>
> We thank the reviewer for pointing out this omission in the initial version. We have now added a Broader Impact section (**Section 7**) discussing the potential implications of improved OOD detection for NLP and multimodal document systems. We highlight that detecting distributional shifts can help reduce overconfident predictions on unfamiliar inputs and improve monitoring of deployed models in applications such as document processing and text classification.
>
> We also discuss limitations of the current study, including the fact that experiments were conducted only on two representative architectures (BERT and LayoutLMv3). While VECO itself is architecture-agnostic and depends only on the geometry of learned representations, broader evaluation across additional models and datasets remains an important direction for future work.
>
> ## References:
>
> [1] Papyan et al.: Prevalence of neural collapse during the terminal phase of deep learning training. PNAS, 2020.
>
> [2] Ben Ammar et al.: NECO: NEural Collapse Based Out-of-distribution detection. ICLR, 2024.
>
> [3] Constantinou et al.: Out-of-Distribution Detection with Attention Head Masking for Multimodal Document Classification. Scientific Reports 2026

---

### Review · Reviewer_9rbr · 2026-02-25

**Summary Of Contributions:**

This work focus on the problem occuring when transferring the post-hoc OOD detection methods into the natural language processing and multimodal document domain.
The main contributions are as follows:
1. It provides the systematic empirical analysis of OOD signals in text and multimodal Transformers. Through extensive experiments on BERT (SST-2 with near/far OOD) and LayoutLMv3 (FinanceDocs, Tobacco3482), the authors show that purely feature-based scores consistently outperform logit-based and hybrid methods. Moreover, common hybrid approaches (e.g., ViM, NECO) often underperform their own feature-only counterparts, indicating that feature-space geometry already provides the primary OOD separability cue in these modalities.

2. This work proposes the VECO, a geometry-aware, purely feature-based OOD scoring framework, which computes a soft contrast between principal-subspace conformity (ID-aligned signal) and residual-subspace deviation (OOD-sensitive signal), avoiding any reliance on classifier logits and the instability of logit–feature fusion in these settings.

3. This work conducts extensive experiments on various benchmarks. The performance indicates the remarkable effectiveness of the VECO.

**Audience:**

Yes

**Audience Explanation:**

Great finding in OOD task in NLP.

**Broader Impact Concerns:**

no ethical issue.

**Claims And Evidence:**

Yes

**Claims Explanation:**

One the one hand, this work provides  experiments on empirical analysis of OOD signals in NLP area indicating an important research direction.

On the other hand, this autho propose comprensvie experiments on the VECO method and the SOTA indicates its effectivenss of method design.

**Requested Changes:**

1. The lack of ablation study.
VECO depends on: 1. the choice of k (principal subspace size), 2. the choice of residual subspace size m, and the scaling factor $\alpha$. However, there is not any efficient ablation study on it, making the method less solid.
2.  VECO proposes a soft-contrast method while the novelty of the method needs verification.
3. The generalizaion ablity of VECO. There are various of text transformers except BERT. Would this method also works on these models is questionaire.

The font size of tables are not aligned such as Table 3 and Table 4. If it exceeds the page limit, then some of the basic baselines can be removed.
The problem also occurs in the Figure 3, there are two dot sub figures while the bar chart is in the middle making it less visually elegant.

---

> ### Author Response · Authors · 2026-03-09
>
> We thank the reviewer for the careful reading of the paper and for the positive evaluation of our work. We appreciate the recognition of the empirical analysis of OOD signals in text and multimodal Transformers, as well as the assessment that VECO achieves strong performance across the evaluated benchmarks.
>
> The reviewer raised several helpful suggestions regarding ablation studies, clarification of the method’s novelty, and additional evaluations. We address these points below. **Additions are highlighted in blue in the revised manuscript**.
>
> ## Ablation for $k, m,$ and $\alpha$
>
> We clarify below both the **design motivation and the empirical stability** observed in our experiments. This was briefly addressed in Appendix A.2, and now is treated comprehensively in **Appendix C.**
>
> ### Choice of $k$ and $m.$
>
> In Section 4 and Appendix A, VECO is motivated by a principal–residual contrast in the feature space. In practice, k is chosen in a representation-geometry-informed manner to capture the dominant ID structure.
>
> These choices reflect the difference in each task representation geometry in our setting: document features exhibit strong low-rank concentration (Figure 1). The first principal component captures the dominant low-rank structure associated with class separation (consistent with NC-like geometry). BERT embeddings are highly anisotropic [3], as can be seen in Figure 6. Retaining components explaining most of the variance ensures that the dominant ID structure is preserved.
>
> We use m = 512 lowest-variance PCA directions across all experiments, following prior residual-based methods (e.g., ViM [4]).
>
> The rationale is geometric:
>
> * Principal directions maximize ID variance
> * Residual directions capture dispersion away from ID structure
> * OOD samples tend to exhibit stronger residual activation
>
> Thus, VECO leverages a structural contrast rather than narrow tuning.
>
> Importantly, we observed in the added **ablation experiments (Figures 4–7) that VECO is not highly sensitive to k or m** within a wide range; performance variations remain small compared to differences across scoring families.
>
> ### Scaling factor $\alpha$.
>
> $\alpha$ is computed from ID training data only, is not tuned on OOD and simply calibrates relative magnitudes of principal and residual energies. This ensures that both signals operate on comparable scales, so that VECO isn’t dominated by whichever subspace has larger raw energy. This is reflected in the added **Table 5**, where this scaling slightly improves and stabilizes performance.
>
> Overall, VECO does not require hyperparameter search on OOD data.
>
> ## Novelty of the soft-contrast formulation
>
> Our contribution is not the observation of principal/residual subspaces per se. Rather:
>
> * We show empirically that in text and multimodal Transformers, **feature-based scores consistently outperform logit-based and hybrid methods**, and hybridization (e.g., ViM [4], NECO [2]) often underperforms their feature-only components. This contrasts with common behavior reported in vision benchmarks and highlights a modality-dependent OOD behavior.
>
> * We propose a **feature-based soft contrast** between ID conformity and residual deviation, motivated by an idealized block-structured Gaussian example of the observed representation geometries, as is illustrated through toy experiments in Section 3.4.
>
> * We propose a **modality-dependent instantiation** (VECO-P vs VECO-M), showing that different conformity signals align better with different representation geometries. VECO unifies geometry-aware signals into a stable framework tailored to the observed constraints.
>
> ## Generalization of VECO
>
> VECO operates on frozen penultimate-layer features and does not depend on architecture-specific components. Its effectiveness relies on representation geometry exhibiting concentration along dominant directions and dispersion in complementary subspaces, a behavior commonly observed in modern deep representations and related to Neural-Collapse-like geometry [1,2]. While we evaluate one representative model per-modality, similar to [5], we expect the framework to extend to other architectures exhibiting similar geometries. We agree that broader architectural evaluation would further strengthen empirical scope and will clarify this.
>
> ## 4. Formatting issues
>
> We thank the reviewer for pointing this out, we will improve Figure 3 layout.
>
> ## References:
> [1] Papyan et al.: Prevalence of neural collapse during the terminal phase of deep learning training. PNAS 2020
>
> [2] Ben Ammar et al.: NECO: NEural Collapse Based Out-of-distribution detection. ICLR 2024
>
> [3] Ethayarajh: How contextual are contextualized word representations? comparing the geometry of BERT, ELMo, and GPT-2 embeddings. NeurIPS 2019
>
> [4] Wang et al.: Vim: Out-of-distribution with virtual-logit matching. CVPR 2022
>
> [5] Constantinou et al.: Out-of-Distribution Detection with Attention Head Masking for Multimodal Document Classification. Scientific Reports 2026

---

### Review · Reviewer_MuvQ · 2026-03-02

**Summary Of Contributions:**

This paper proposes VECO,an OOD detection method. VECO operated by performing PCA on the embedding space of in-distribution data, and for a new query, the norm of both the top k and residual components are computed. The intuition is that in-distribution queries should achieve large top k norms, whereas OOD data should achieve large residual norms. Thus, VECO combines these two norms into a single score, which is increasing in the top k norm, and decreasing on the residual norm (the authors actually propose two such scores combining the two norms). The authors test VECO on document and text OOD detection, achieving good empirical results.

Overall, the paper is easy to follow and well written, although I believe a few relevant details are missing (see requested changes).

**Audience:**

Yes

**Audience Explanation:**

Yes, OOD detection is of clear interest to a subset of TMLR's community, as are the findings and proposed method.

**Claims And Evidence:**

Yes

**Claims Explanation:**

I believe the claims made in the paper are backed by evidence.

The authors motivate VECO through a simplified Gaussian example, where their score corresponds exactly to a likelihood ratio. I believe this is a good motivation for their proposed score.

Empirically, the testing performed by the authors is quite thorough, and I also believe that it presents convincing evidence supporting VECO's performance. I do nonetheless have some uncertainty about this claim, as I am much more familiar with the OOD detection literature in CV than in NLP, so it is possible that there's a relevant baseline I am unaware of that the authors omitted.

**Requested Changes:**

1. I think the organization of the paper is odd: having exxperimental settings -> main method -> experiments breaks the flow. I think moving section 3 after presenting VECO would improve the flow of the paper.

2. Although related work and the used baselines are cited, they are not discussed in enough detail. Pretty much the only details currently given are whether they are based on representations, logits, or both. Describing the baselines in more detail, specially the ones based on representations (like VECO), would be valuable to have more clarity as to how VECO differs from existing work.

3. There's a very brief discussion in the appendix about the choice of $k$. What about the choice of $m$, how did you choose this hyperparameter? This should also be specified in the paper.

4. The notation in equation 3 should be changed: as written as an expectation over in-distribution data, $U_k$ would be random as well, whereas I am sure that's only computed once and fixed. Instead, I think you should use a summation over $i$ rather than an expectation.

5. The values $\bar{w}$ and $\gamma_r$ used in equation 5 should be clearly defined in the main text. Additionally, the value of $\alpha$ defined in equation 3 comes out of nowhere, and I believe that a discussion about how $\alpha$ relates to $\bar{w}$ and $\gamma_r$ in the idealized Gaussian example could be insightful.

6. The sentence "the induced ranking is a monotone function of the pair $(s_p, s_r)$" is very unclear. What does it mean for a function to be monotonic on a pair of inputs? Please rephrase.

7. You introduced two versions of VECO (equations 8 and 9). Do you keep hyperparameters (choice of $\alpha$, k, and $m$) the same for both versions? Please specify so in the main text as well.

---

> ### Author Response · Authors · 2026-03-09
>
> We thank the reviewer for the careful reading of the paper and for the positive assessment of our work. We are pleased that the reviewer found the paper easy to follow and well written, and that the empirical evaluation was considered thorough and convincing. We also appreciate the recognition that the Gaussian example provides a useful motivation for the proposed score.
>
> The reviewer raised several helpful suggestions aimed at improving clarity and completeness (paper organization, baseline descriptions, notation, and hyperparameter details). We have incorporated these suggestions in the revised manuscript and detail the corresponding changes and clarifications below.
>
> **All the additions made to the paper will be highlighted in blue**, and will be referred to precisely in the following.
>
> ## Organization, Baselines description, and monotonicity clarification
>
> We thank the reviewer for pointing this out. We agree that presenting the experimental setting before introducing VECO disrupts the flow. In the revised version, we move Section 3 to follow the presentation of VECO.
>
> We also expanded the description of the baselines in **Appendix A** to better clarify their mechanisms and how they differ from VECO.
>
> Finally, we agree that the monotonicity phrasing was unclear. We now explicitly state that the score is strictly increasing in $s_p$ and strictly decreasing in $s_r$.
>
> ## Hyperparameters: choice and sharing across variants
>
> Equations (2) and (9) define VECO-P and VECO-M, respectively. VECO-M does not depend on $k$, as it replaces the principal energy with a Mahalanobis conformity term.
>
> Across all experiments, the residual dimension is fixed to $m = 512$ and is shared across both variants, following prior residual-based methods (e.g., ViM [1]) with added experiments (**Appendix C**) showing minimal sensitivity around this range.
>
> The principal dimension $k$ is chosen in a representation-geometry-informed manner to capture the dominant ID structure rather than tuned for OOD performance. These choices reflect the difference in each task representation geometry in our setting: document features exhibit strong low-rank concentration (Figure 1), while BERT embeddings are highly anisotropic [2].
>
> The rationale is geometric:
>
> * Principal directions maximize ID variance
> * Residual directions capture dispersion away from ID structure
> * OOD samples tend to exhibit stronger residual activation
>
> Thus, VECO leverages a structural contrast rather than narrow tuning.
>
> We have added a section, **Appendix C**, to highlight VECO stability to these hyperparameters. Overall, VECO does not require hyperparameter search on OOD data.
>
> ##  Equation 3 expectation notation
>
> We agree that the expectation notation may suggest that $\mathbf{U}_k$ is random. We replace the expectation with an empirical summation over the ID training set, clarifying that the PCA basis is computed once and fixed at test time.
>
> ## Clarification of $\bar{w}$ or $\gamma_r$, and $\alpha$
>
> We thank the reviewer for pointing out that $\bar{w}$ or $\gamma_r$ were only defined in the appendix and that the role of $\alpha$ was not clearly discussed in relation to the Gaussian example. Importantly, as clarified in the “VECO as a smooth contrast” paragraph, VECO does not attempt to approximate the likelihood ratio itself, but rather the induced ordering by depending on the same geometric quantities $(s_p, s_r)$ with the same monotonic directions (strictly increasing in $s_p$ and strictly decreasing in $s_r$).
>
> The scalar $\alpha$ therefore serves only as a calibration constant computed from training data to normalize the relative magnitudes of principal and residual energies. This prevents the soft contrast from being dominated by whichever subspace exhibits larger raw energy.
>
> We now emphasize this interpretation in the main text and added a remark after Equation (5) detailing the expressions and interpretations of $\bar{w}$ and $\gamma_r$. **Table 5** further illustrates that this calibration improves stability.
>
> ## Completeness of evaluated baselines
>
> We appreciate the reviewer’s comment. Our benchmark includes the baselines most commonly used for post-hoc OOD detection in NLP and multimodal settings, including logit-based scores (MSP, MaxLogit, Energy), feature-space methods (Mahalanobis, Residual, NaN), and hybrid approaches combining logits and representations (ViM, ASH, NECO). To improve clarity, we expanded the description of these methods in the revised manuscript (Appendix A).
>
> ## References:
>
> [1] Wang et al.: Vim: Out-of-distribution with virtual-logit matching. CVPR, 2022.
>
> [2] Ethayarajh: How contextual are contextualized word representations? comparing the geometry of BERT, ELMo, and GPT-2 embeddings. NeurIPS, 2019.

---

> > ### Comment · Reviewer_MuvQ · 2026-03-18
> > **Thank you**
> >
> > Thank you for your response, all my concerns have been addressed. As a small final point, I just noticed that table 4 is slightly too large and extend beyond its caption. I'd recommend fixing this in the camera-ready either by making the font smaller, or by splitting the table into 2 tables.

---

### Decision · Action_Editor_zvxZ · 2026-04-24

**Recommendation:** Accept as is

**Additional Comments:**

This paper proposes a new out-of-distribution (OOD) detection method named VECO (VEctor COnformity). The main claim is that most existing approaches are post-hoc OOD detection methods that are designed for vision models and thus achieve suboptimal performance on text and multimodal models. To address this, the paper proposes a geometry-aware, purely feature-based OOD scoring framework, showing superior performance. The analysis also highlights the modality-dependent nature of OOD detection.

There are several strengths. The paper is well written and easy to follow. The organization and narrative are good. The motivation and the claim are clear. It first reveals that existing post-hoc OOD detection methods are designed for vision models and are thus suboptimal for text and multimodal models. Based on this observation, it proposes a feature-based framework and achieves improvement. The finding of the modality-dependent nature of OOD detection is also novel. Overall, this paper provides a new perspective and valuable insights for OOD detection across different modalities. The experimental results demonstrate that the proposed method achieves state-of-the-art performance.

**Audience:**

Yes

**Audience Explanation:**

The audiences from the OOD domain would be interested in this work.

**Claims And Evidence:**

Yes

**Claims Explanation:**

The claims made in the paper are supported by evidence. The authors motivate VECO through a simplified Gaussian example, where their score corresponds exactly to a likelihood ratio. I believe this is a good motivation for their proposed score. Empirically, the testing performed by the authors is quite thorough, and I also believe that it presents convincing evidence supporting VECO's performance.